# Functional ultrasound imaging of stroke in awake rats

**Clément Brunner[1,2,3,4], Gabriel Montaldo[1,2,3,4], Alan Urban[1,2,3,4]\***

[1]Neuro-Electronics Research Flanders, Leuven, Belgium; [2]Vlaams Instituut voor Biotechnologie, Leuven, Belgium; [3]Interuniversity Microelectronics Centre, Leuven, Belgium; [4]Department of Neurosciences, KU Leuven, Leuven, Belgium

**\*For correspondence:**
alan.urban@nerf.be

**Competing interest:** The authors declare that no competing interests exist.

**Abstract** Anesthesia is a major confounding factor in preclinical stroke research as stroke rarely occurs in sedated patients. Moreover, anesthesia affects both brain functions and the stroke outcome acting as neurotoxic or protective agents. So far, no approaches were well suited to induce stroke while imaging hemodynamics along with simultaneous large-scale recording of brain functions in awake animals. For this reason, the first critical hours following the stroke insult and associated functional alteration remain poorly understood. Here, we present a strategy to investigate both stroke hemodynamics and stroke-induced functional alterations without the confounding effect of anesthesia, i.e., under awake condition. Functional ultrasound (fUS) imaging was used to continuously monitor variations in cerebral blood volume (CBV) in +65 brain regions/hemispheres for up to 3 hr after stroke onset. The focal cortical ischemia was induced using a chemo-thrombotic agent suited for permanent middle cerebral artery occlusion in awake rats and followed by ipsi- and contralesional whiskers stimulation to investigate on the dynamic of the thalamocortical functions. Early (0–3 hr) and delayed (day 5) fUS recording enabled to characterize the features of the ischemia (location, CBV loss), spreading depolarizations (occurrence, amplitude) and functional alteration of the somatosensory thalamocortical circuits. Post-stroke thalamocortical functions were affected at both early and later time points (0–3 hr and 5 days) after stroke. Overall, our procedure facilitates early, continuous, and chronic assessments of hemodynamics and cerebral functions. When integrated with stroke studies or other pathological analyses, this approach seeks to enhance our comprehension of physiopathologies towards the development of pertinent therapeutic interventions.

## eLife assessment

This **important** proof-of-concept study strongly supports the utility of functional ultrasound imaging for evaluating cerebral hemodynamics in rat models of brain injury. Functional ultrasound affords a distinct coverage/spatial/temporal resolution tradeoff when compared to other modalities for studying brain hemodynamics. The **solid** data presented indicate high fidelity of the recordings, a particular feat given that the rats were awake. On the other hand, single slice imaging and complexity of registration of subsequent imaging sessions limit the usefulness of the approach, particularly for quantitative imaging, and the small sample size will need to be followed up with and verified by future studies. This work will be of interest to researchers working in functional neuroimaging and more precisely with preclinical models of stroke in rodents.

## Introduction

Stroke is a multifaceted and multiphasic pathology, complex to mimic under experimental conditions. Indeed, when compared to clinics, preclinical stroke models suffer from several limitations that

narrow the experimental focus on a few conditions (*Macrae, 2011*; *Fluri et al., 2015*; *Sommer, 2017*). Among these limitations, one can highlight the complexity to combine (i) imaging stroke in conscious animal models, (ii) addressing post-stroke brain functions, and (iii) recording of hyperacute stroke hemodynamics, all crucial to design timely effective therapeutic strategies.

As the first limitation, the use of anesthesia in preclinical studies seems to hamper the transition from animal to patient as most of stroke occurs in awake or sleeping patients (*Mackey et al., 2011*; *Muir, 2023*), but rarely in sedated patients. Moreover, anesthetics disrupt the brain functionality, alters neurovascular coupling (*Reimann and Niendorf, 2020*; *Masamoto and Kanno, 2012*), while differentially affecting metabolism, electrophysiology, temperature, blood pressure, and tissue outcome by acting as neurotoxic or neuroprotective agents (see reviews *Traystman, 2010*; *Hoffmann et al., 2016*; *Slupe and Kirsch, 2018*).

To date, only a few groups succeeded in inducing a stroke in awake rodents (*Seto et al., 2014*; *Lu et al., 2014*; *Balbi et al., 2017*; *Sunil et al., 2020*). Moreover, post-stroke network and functional alterations have been addressed by few preclinical studies, providing evidence of functional network reorganization from minutes (*Mohajerani et al., 2011*; *Brunner et al., 2018*) to days *Dijkhuizen et al., 2001*; *Dijkhuizen et al., 2003*; *Abo et al., 2001*; *Weber et al., 2008*; *Shih et al., 2014* following stroke onset. However, these studies mostly focused on the cortical readouts and were unable to capture how deeper brain regions, like thalamic relays, were functionally and/or temporally affected remotely from the stroke insult (e.g. diaschisis) (*Zhang et al., 2012*; *Carrera and Tononi, 2014*; *Cao et al., 2020*). Furthermore, these studies were always conducted using various anesthetics (e.g. ventilated with halothane or isoflurane; medetomidine, urethane) known to differentially impact brain functions, as mentioned above.

Until recently, live monitoring of the hyperacute stroke-induced hemodynamics was restricted to few methods but often focused to the brain surface (*Balbi et al., 2017*; *Levy et al., 2012*; *Dunn, 2012*). On the other hand, fUS, a recent neuroimaging modality measuring cerebral blood volume changes (CBV) (*Macé et al., 2011*; *Demené et al., 2019*; *Montaldo et al., 2022*), was successfully employed to measure brain functions of awake rodents (*Urban et al., 2015*; *Sieu et al., 2015*; *Macé et al., 2018*; *Bergel et al., 2020*; *Brunner et al., 2020*; *Brunner et al., 2021*), to address early post-stroke functional reorganization (*Brunner et al., 2018*), and to track stroke-induced hemodynamics at the brain-wide scale (i.e. ischemia and spreading depolarization *Brunner et al., 2023*). However,

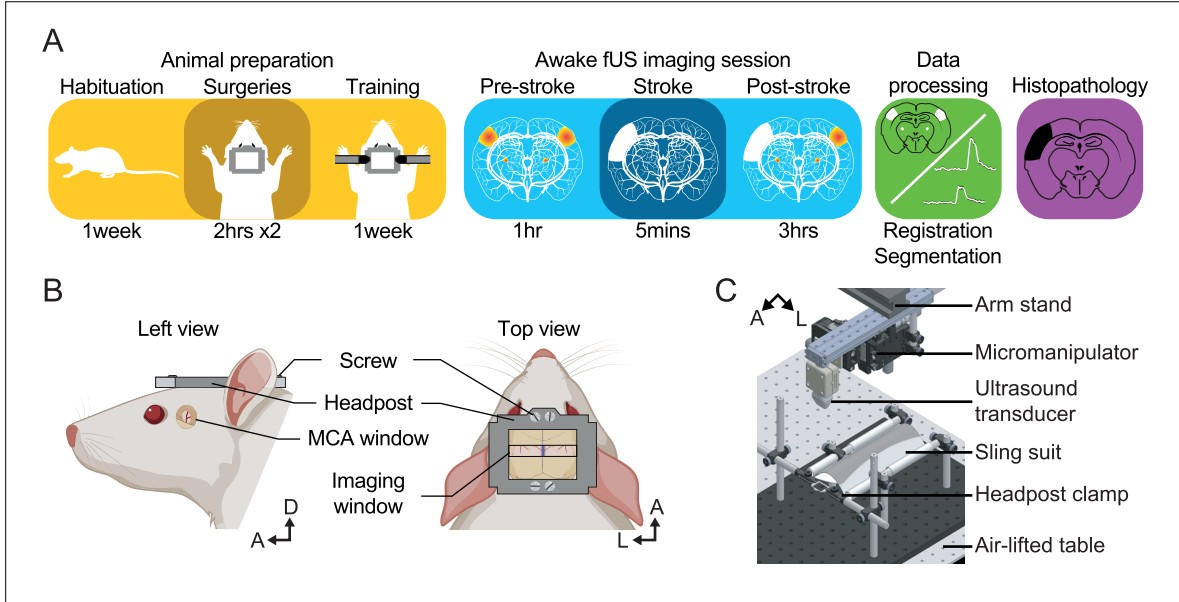

**Figure 1.** Experimental procedure. (**A**) Workflow for brain imaging of awake head-fixed rats including, from left to right: animal preparation (habituation to the bench, implantation of cranial windows, training), functional ultrasound (fUS) imaging of stroke induction and brain functions, data processing, and histopathology. (**B**) Overview of the headpost implantation and cranial windows developed for combined MCAo (left) and brain imaging (right) under awake conditions. (**C**) Computer-aided design of the experimental apparatus where the animal is placed and secured in a suspended sling suit and the head fixed by the means of clamps holding the headpost implanted to the rat skull. A, Anterior; D, Dorsal; L, Left.

no study has further exploited such strategies to combine stroke hemodynamics and brain-wide functional alteration in awake rodents.

In this study, we report on the stroke induction and the alteration of somatosensory brain functions in awake rats. Using the latest improvements toward imaging of awake rodents (*Urban et al., 2015*; *Macé et al., 2018*; *Brunner et al., 2020*) combined with chemo-thrombotic agent directly applied to the middle cerebral artery (MCA) (*Karatas et al., 2011*; *Syeara et al., 2020*), we were able to induce MCA occlusion (MCAo) in awake rats while capturing continuous hemodynamic changes, including ischemia and spreading depolarization, in +65 brain regions for up to 3 hr after stroke onset. Finally, we investigated on how somatosensory thalamocortical functional responses were progressively altered from early (0–3 hr) to late post-stroke (5d) timepoints.

## Results

### Animals

Report on animal use, experimentation, and exclusion criteria can be found in *Supplementary file 1*. Rat #1 was excluded after the control session as the imaging window was too anterior to capture both cortical and thalamic responses. Rat #2 was excluded as hemodynamic responses were inconsistent during baseline (pre-stroke) period. Rat #9 showed early post-stroke reperfusion and was excluded from stroke analysis, the control session (pre-stroke) from Rat #9 was analyzed. All imaging sessions started at approximately 8 am, aligning closely with the end of their active phase.

### Real-time imaging of stroke induction in awake rats

We first developed a dedicated procedure for real-time imaging of stroke induction and associated evoked functional deficits in awake head-fixed rats (*Figure 1A*). Each rat was subjected to two cranial windows accessing independently the distal branch of the left MCA (*Figure 1B*, Left) and the selected brain regions to image (*Figure 1B*, Right). The latter was performed between bregma –2 and –4 mm allowing for jointly monitoring the bilateral thalamocortical circuits of the somatosensory whisker-to-barrel pathway, including the ventroposterior medial nucleus of the thalamus (VPM) and the primary somatosensory barrel-field cortex (S1BF). Moreover, the selected coronal cross-section includes the posterior nucleus of the thalamus (Po), the reticular nucleus of the thalamus, and the ventral part of the zona incerta are known for relaying information related to whiskers (*Adibi, 2019*; *Bosman et al., 2011*), and also direct efferent projections from the S1BF to other cortical and subcortical regions (*Zakiewicz et al., 2014*). Prior to imaging sessions, rats were extensively trained to accept comfortable restraints in the experimental apparatus (*Figure 1C*), suitable for fUS recording of brain functions and stroke induction under awake conditions. After data acquisition, the coronal cross-section was registered and segmented on a custom-developed digital rat atlas (*Brunner et al., 2022a*) to provide a dynamic view of the changes in perfusion induced either by the stroke or evoked activity.

To overcome the limitations of conventional stroke models, we occluded the distal branch of the MCA by the mean of a chemo-thrombotic ferric chloride solution (FeCl$_3$) (*Karatas et al., 2011*; *Syeara et al., 2020*) while performing fUS imaging in awake rats (*Figure 2A*). It should be mentioned that the rats did not show any obvious signs of pain or discomfort (e.g. vocalization, aggressiveness) during the restrain period and occlusion procedure. The MCA occlusion (MCAo) was captured live with fUS and confirmed by the large drop of signal, i.e., ischemia, localized in the cortex of the left hemisphere (*Figure 2B and C*, *Video 1* and *Figure 2—figure supplement 1*) as shown with μDoppler image taken 3 hr and 5d after the stroke onset (dashed outline, *Figure 2B*, Top row). Bmode images accounting for the brain tissue echogenicity remain unchanged early after stroke onset (3 hr) while showing focal hyper-echogenicity (dashed outline, *Figure 2B*, Bottom row) lately after stroke onset (5d) as a marker of focal lesion (*Gómez-de Frutos et al., 2021*). The stroke-induced hemodynamic changes have been continuously recorded for up to 3 hr after stroke onset, registered and segmented into 69 regions (*Figure 2—figure supplement 1*). We first extracted the average change in rCBV (ΔrCBV in %) in the S1BF cortex of the left hemisphere (blue region, *Figure 2B*) and detected an abrupt drop of rCBV down to ~40% of the baseline level after the occlusion of the MCA, followed by a progressive decrease of the rCBV to 30% of baseline level 3 hr after the stroke onset (*Figure 2C* and *Video 1*). Second, we extracted the average rCBV change from a cortical region supplied by the anterior cerebral artery directly after the MCAo. The signal extracted from the retrosplenial granular cortex (RSGc;

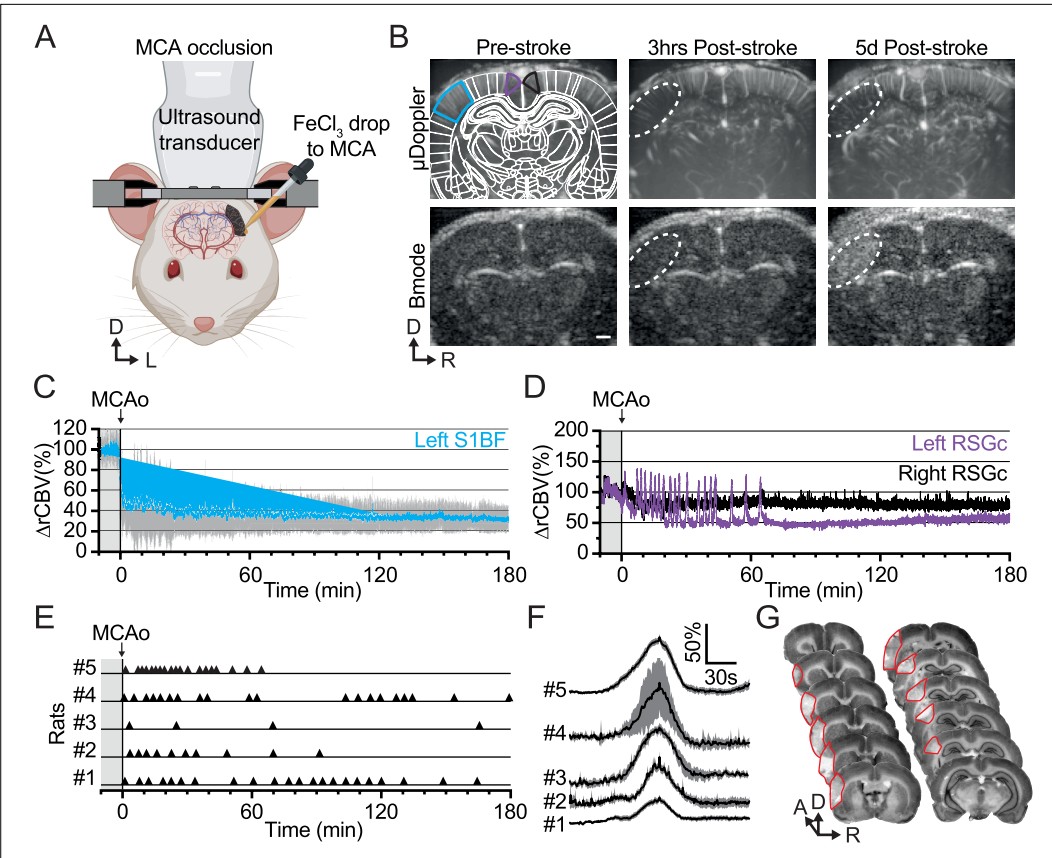

**Figure 2.** Ferric chloride (FeCl₃)-stroke induction under awake conditions. (**A**) Front view representation of functional ultrasound (fUS) imaging during live chemo-thrombosis of the left middle cerebral artery (MCA) with FeCl₃ in awake head-fixed rats. (**B**) Set of typical coronal μDoppler images of the brain microvasculature (top row) and morphological Bmode images (bottom row) before stroke (left), 3 hr (middle), and 5d after stroke onset (right) from the same animal. μDoppler images (top left) were registered and segmented based on a digital version of the rat brain atlas (white outlines). Colored outlines (cyan, purple, and black) delineate regions of interest plotted in (**C**) and (**D**). The white dotted region of interest highlights the ischemia in μDoppler images (Top row) and tissue hyper-echogenicity in Bmode (Bottom row). (**C**) Temporal plot of the average signal (ΔrCBV (%), mean ± 95% CI, n=5) in the barrel-field primary somatosensory cortex (S1BF, cyan) from the left hemisphere, affected by the MCA occlusion (MCAo). (**D**) Temporal plots of the average signal (ΔrCBV (%)) in the retrosplenial granular cortex (RSGc) from the affected (purple) and non-affected hemisphere (black) from the same animal. (**E**) Occurrence of spreading depolarizations after MCAo. Each horizontal line represents one rat; each triangle marker depicts the occurrence of one spreading depolarization. (**F**) Temporal plots of the average signal change (ΔrCBV (%), mean ± 95% CI, respectively black line and gray band) of hemodynamic events associated with spreading depolarizations (centered on the peak) for each rat (#1–5). (**G**) Typical rat brain cross-sections stained by cresyl violet to evaluate the tissue infarction at 24 hr after FeCl₃-induction occlusion of MCA. The infarcted territory is delineated in red. Scale bars: 1 mm. D: Dorsal; L: left; R: right.

The online version of this article includes the following figure supplement(s) for figure 2:

**Figure supplement 1.** Hemodynamic changes (relative cerebral blood volume (rCBV) in %) induced by MCA occlusion (MCAo) in 69 regions located in the ipsilesional (left panel) and contralesional hemisphere (right panel) of the imaged coronal cross-section.

purple and black regions in *Figure 2B*) shows successive and transient increases of signal. It characterizes hemodynamic events associated with spreading depolarizations (SDs) in the left hemisphere (in purple; *Figure 2D* and *Video 1*) while resulting in a slight and stable oligemia in the right hemisphere (in black; *Figure 2D* and *Figure 2—figure supplement 1*). SD events were observed in the peri-ischemic territory of all rats subjected to MCAo and occurred in an ostensibly random fashion (*Figure 2E*); however, hemodynamic events associated with SDs showed a similar bell shape and timecourse across animals (*Figure 2F*). On average, we detected five SD events per hour per rat. Finally,

**Video 1.** Movie of hemodynamic changes induced by middle cerebral artery (MCA) occlusion using ferric chloride (FeCl₃) in awake head-fixed rats. Raw images. https://elifesciences.org/articles/88919/figures#video1

we stained brain slices 24 hr after MCAo and confirmed that FeCl₃-induced ischemia turned into tissue infarction (red delineation; *Figure 2G*).

## Stroke-induced alterations of the thalamocortical functions

One hour before and during 3 hr after the occlusion of the MCA, rats received mechanical stimulation of the whisker alternately delivered to the left and right pad using motorized combs (5 Hz sinusoidal deflection, 20° amplitude, 5 s duration; *Figure 3A*) to capture the spatiotemporal dynamics of the functional circuit. Before stroke, the sensory-evoked stimulations elicited a robust and statistically significant functional response (z-score >1.6, see Material and methods) for both left and right stimulation (orange and green, respectively; z-score maps; Pre-stroke panel, *Figure 3B* and *Video 2*) with the activity spatially confined in the contralateral dorsal part of the VPM and S1BF. The temporal analysis of the somatosensory evoked responses in the contralateral hemisphere confirmed that VPM, Po, and S1BF regions were significantly activated and for both left and right stimuli (****p<0.0001, ***p<0.001 and ****p<0.0001, respectively; Left panel, *Figure 3C*). We also detected significant increase of activity in S2, AuD, Ect (****p<0.0001) and PRh (***p<0.001) cortices and VPL nucleus (**p<0.01; the list of acronyms is provided in *Supplementary file 2*), brain regions receiving direct efferent projections from the S1BF (*Zakiewicz et al., 2014*; *Fabri and Burton, 1991*; *Frostig et al., 2008*), VPM or Po nuclei (*Viaene et al., 2011*; *El-Boustani et al., 2020*; *Landisman and Connors, 2007*). It is worth noted that no habituation or sensitization due to the repetitiveness of whiskers stimulation was observed in cortical and subcortical regions over the pre-stroke sessions (*Figure 3—figure supplement 1*).

After the stroke, the activity map from the left pad stimulation elicited a similar response pattern as pre-stroke; however, the right pad stimulation showed a total absence of functional response in the S1BF cortex and a significant reduction of the response in the VPM (z-score maps; Post-stroke panel, *Figure 3B*, and *Video 2*). Over the 3 hr following stroke onset, functional responses to left whisker stimulation were still detected in the cortical and thalamic regions of the contralateral (right) hemisphere; however, functional responses to right whisker stimulation were only detected in subcortical nuclei (i.e. VPM, Po, VPL), while attenuated when compared with the responses from the pre-stroke period and from the other hemisphere (*Figure 3B and C*). Furthermore, no responses were detected at the cortical level (S1BF, S2, and AuD; right panel, *Figure 3B and C*). A larger version of *Figure 3C* is provided in *Figure 3—figure supplement 2*.

To better evaluate how the functional responses were affected by the stroke, we have divided the post-stroke recording period into three sections of 1 hr each and compared them with the 1 hr pre-stroke period (*Figure 3D*). Temporal plots from the pre-stroke period showed robust increases in signal during the stimulus in S1BF, VPM, and Po regions and high consistency between left and right stimuli (black plots, *Figure 3D* and *Figure 3—figure supplements 2–3*); fitting well the hemodynamic response functions as previously observed (*Brunner et al., 2018*; *Hirano et al., 2011*). Indeed, the hemodynamic responses were characterized by a quick increase in signal during whisker stimulation reaching a peak after 4.0 s at 18.2 ± 1.3% (4.0 s, 18.6 ± 1.2%) of baseline level for S1BF, 4.0 s at 4.6 ± 0.5% (3.2 s, 5.8 ± 0.7%) for VPM, and 2.4 s at 2.9 ± 0.7% (3.2 s, 4.0 ± 0.8%) for Po from the left stimulation (right, respectively; mean ±95% CI) before slowly returning to baseline level (black plots, *Figure 3D*).

During the first hour following the stroke onset, functional responses in the left hemisphere (i.e. ipsilesional) were abolished in the S1BF, S2, and AuD (0–1 hr Post-stroke, ****p-value <0.0001), significantly decreased in the VPM (0–1 hr Post-stroke, ***p-value <0.001), and unchanged in Po and VPL (0–1 hr Post-stroke, ns; *Figure 3D*) when compared with the pre-stroke responses (Pre-stroke, black plots, *Figure 3D*). Over the two following hours (i.e. 1–2 hr and 2–3 hr Post-stroke), the hemodynamic

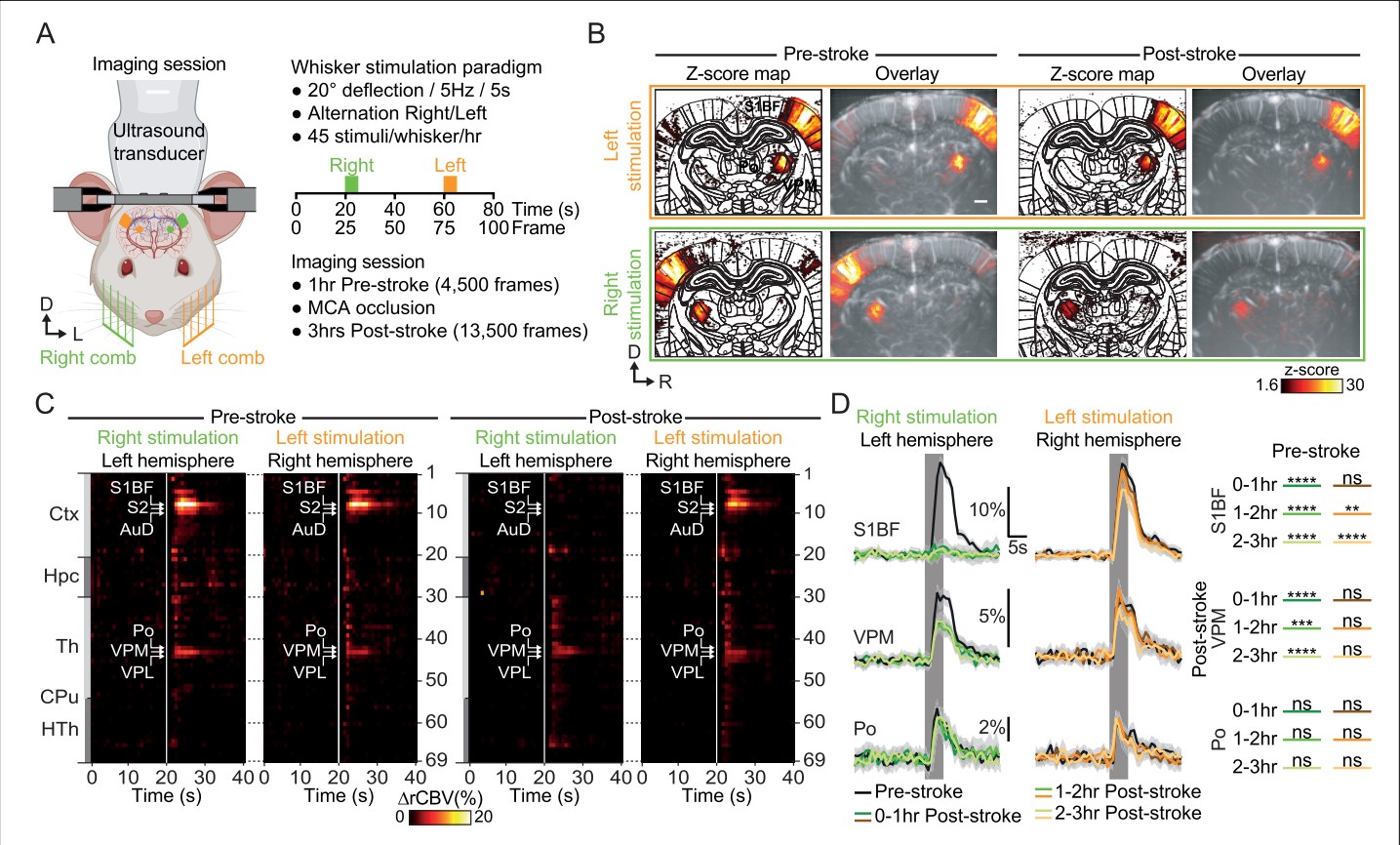

**Figure 3.** Early post-stroke alteration of whisker-to-barrel thalamocortical circuit. (**A**) Front view representation of functional ultrasound (fUS) imaging during repetitive stimulation of the left (orange) or right whisker pad (green) with a mechanical comb in awake head-fixed rats. Whisker stimulations were delivered alternately between left and right whisker pads before and early after MCA occlusion (MCAo). Each rat receives 45 stimuli per whisker pad each hour of imaging. (**B**) Average activity maps (z-score) from one rat depicting evoked functional responses to either left (orange) or right whisker pads stimulation (green) registered with a digital version of the rat Paxinos atlas (white outlines) and overlaid with the corresponding coronal μDoppler image, before (left; Pre-stroke, average of 45 trials) and after stroke induction in the left hemisphere (right; Post-stroke, average of 125 trials). (**C**) Region-time traces of the average hemodynamic changes (ΔrCBV (%)) in response to right (green) or left whisker stimulation (orange) extracted from the contralateral hemisphere (left and right, respectively) before (left; Pre-stroke, n=5, 45 trials/rat) and after stroke induction in the left hemisphere (right; Post-stroke, n=5, 135 trials/rat). Brain regions are ordered by major anatomical structures (see *Supplementary file 2*). The vertical line represents the stimulus start. S1BF, S2, AuD, VPM, VPL, and Po regions are brain regions significantly activated (all pvalue <0.01; GLM followed by t-test). A larger version of panel C is provided in *Figure 3—figure supplement 2*. (**D**) Left, Average response curves from the S1BF, the VPM, and Po regions before (Pre-stroke, black, n=5, 45 trials/rat), and from first to third hour after stroke induction (0–1 hr, 1–2 hr, 2–3 hr Post-stroke, orange and green, n=5, 45 trials/hr/rat). Data are mean ± 95% CI. The vertical bar represents the whisker stimulus. Right, Statistical comparison of the area under the curve (AUC) between pre-stroke and post-stroke response curves for S1BF, VPM, and Po regions (Non-parametric Kruskal-Wallis test corrected with Dunn's test for multiple comparisons; ns: non-significant; *p<0.05; **p<0.01; ***p<0.001; ****p<0.0001. See also *Figure 3—figure supplement 3*). Scale bars: 1 mm. D: Dorsal; L: left; R: right; Ctx: Cortex; Hpc: Hippocampus; Th: Thalamus; CPu: Caudate Putamen; HTh: Hypothalamus; S1BF: barrel-field primary somatosensory cortex; S2: Secondary somatosensory cortex; AuD: Dorsal auditory cortex; VPM: Ventral posteromedial nucleus of the thalamus; VPL: Ventral postero-lateral nucleus of the thalamus; Po: Posterior nucleus of the thalamus.

The online version of this article includes the following figure supplement(s) for figure 3:

**Figure supplement 1.** Averaged hemodynamic response curves (ΔrCBV in %) of 45 consecutive right (green) or left whisker stimulation (orange; 1 hr recording) extracted in the contralateral S1BF, VPM, and Po regions (top to bottom).

**Figure supplement 2.** Close-up view of *Figure 3C*.

**Figure supplement 3.** Top Panel – Violin plots showing the distribution of the area under the curve (AUC) extracted from hemodynamic response time-courses of individual trials in S1BF (top row), VPM (middle row), and Po regions (bottom row), for stimulation delivered either to the right (left column) or left whisker pad (right column) along all the periods of the recording (Pre-Stroke, 0–1 hr Post-stroke, 1–2 hr Post-Stroke, 2–3 hr Post-Stroke).

**Figure supplement 4.** Activity maps, region-time traces of the 69 brain regions imaged, mean and individual time-courses for all trials (left and right stimuli - including contra- and ipsilateral traces) and imaging timepoints (Control, Pre-Stroke, Post-Stroke) for all the rats included in this work.

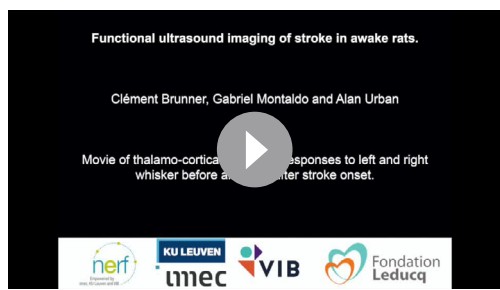

**Video 2.** Movie of thalamocortical functional responses to left and right whisker stimulation before and 3 hr after stroke onset.

https://elifesciences.org/articles/88919/figures#video2

responses captured in these regions remained similar as those detected during the first post-stroke hour (green plots, *Figure 3D*).

Regarding the right hemisphere (i.e. contralesional), the functional responses of S1BF and VPM were conserved during the first hour after the stroke onset (ns, 0–1 hr Post-stroke; orange plots, *Figure 3D*). During the two following hours, signal changes in S1BF show a significant and progressive decrease of activity (1–2 hr Post-stroke **p-value <0.01, 2–3 hr Post-stroke ****p-value <0.0001; orange plots, *Figure 3D*; Similar observations were made for S2 and AuD) whereas responses in VPM remained stable during the second hour post-stroke (1–2 hr, ns) before significantly decreasing during the third hour (2–3 hr Post-stroke *p-value <0.05; orange plots, *Figure 3D*). Finally, the functional responses in VPM and Po remained unchanged over the 3 hr following the stroke onset (bottom panel, *Figure 3D*).

Activity maps, region-time traces of the 69 brain regions, mean and individual time-course for all trials (left and right stimuli - including ipsi and contralateral traces), imaging timepoints (Control, Pre-Stroke, Post-Stroke) for all the rats included in this work can be found in *Figure 3—figure supplement 4*.

## Delayed alteration of the somatosensory thalamocortical pathway

A secondary objective of this work was to evaluate the fUS ability to identify potential delayed functional alteration within a few days after the initial injury. Two animals were imaged five days after the MCAo following the same experimental, stimulation, imaging, and processing conditions as for the early post-stroke session. Given that only two rats were imaged at this timepoint, the findings presented here should be viewed as preliminary or proof of concept. Additional data will be essential for validation. Consequently, no statistical analysis was conducted for this segment of the study.

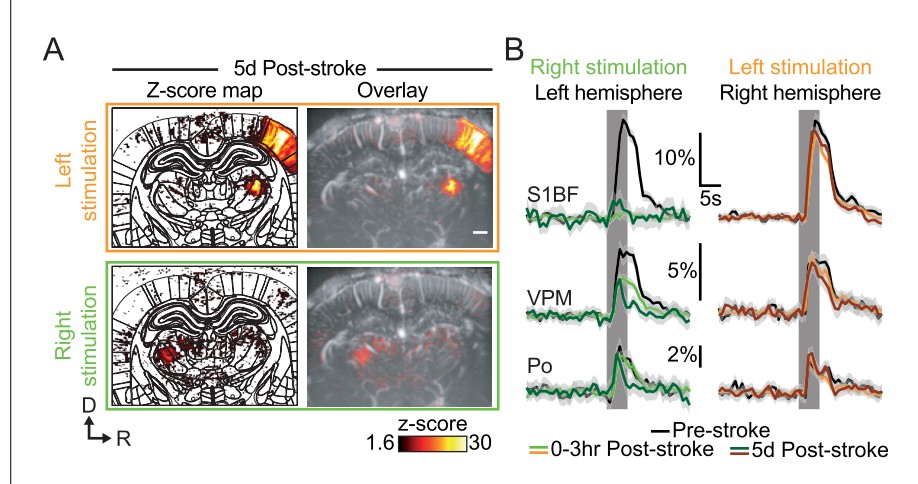

**Figure 4.** Late post-stroke alteration of whisker-to-barrel thalamocortical circuit. (**A**) Activity maps (z-score; average of 45 trials) depicting evoked functional responses to left (orange) or right whisker pads stimulation (green) 5d after stroke induction. Z-score maps are registered with the Paxinos atlas (white outlines; Left) and overlaid with the corresponding coronal μDoppler image (Right). (**B**) Left; Average response curves to left and right whisker stimulation (orange and green; respectively) extracted from S1BF, VPM, and Po before (Pre-stroke, black, n=2, 45 trials/rat), 0–3 hr (0–3 hr Post-Stroke; light orange/green, n=2, 45 trials/hr/rat) and 5d after stroke induction (5d Post-stroke, dark orange/green, n=2, 45 trials/rat). Data are mean ± 95% CI. The vertical bar represents the whisker stimulus. Scale bars: 1 mm. D: Dorsal; R: right; S1BF: barrel-field primary somatosensory cortex; VPM: Ventral posteromedial nucleus of the thalamus; Po: Posterior nucleus of the thalamus.

Activity maps, region-time traces, and individual trials for both right and left stimulation (including ipsi- and contralateral) for each rat are provided in *Figure 3—figure supplements 2–3*.

Five days after the MCA occlusion, we first placed the ultrasound probe over the imaging window and adjusted its position (using micromanipulator) to find back the recording plane from Pre-Stroke session using Bmode (morphological mode) and µDoppler imaging using brain vascular landmarks (i.e. vascular patterns, brain surface, and hippocampus *Brunner et al., 2021*; *Brunner et al., 2023*; see *Figure 2B*). Functional responses to left whisker stimulation were still detected in the right hemisphere (i.e. contralesional), at the cortical and subcortical levels (orange; *Figure 4A*). As for the early post-stroke imaging period, the functional responses to right whisker stimulation were only detected in the subcortical nuclei and not at the cortical level (green; *Figure 4A*).

Second, we extracted and compared the temporal plots of the functional responses gathered 5d after the stroke with the one obtained from the same two animals at the pre-stroke and 3 hr post-stroke timepoints (*Figure 4B*). At this later time point, the functional responses in the left S1BF (dark green plot, left panel, *Figure 4B*. Similar observations were made for the S2 and AuD) remained abolished when compared with the pre-stroke period (black plot), while slightly increased when compared with the 3 hr post-stroke timepoint (green plot). The responses detected in the VPM 5d after the stroke onset (dark green plot, left panel, *Figure 4B*) were largely decreased not only when compared with the pre-stroke signal (black plot) but also with the 3 hr post-stroke trace (green plot). Interestingly, both the amplitude and time-to-peak of the hemodynamic response function were very similar to those from the early post-stroke signal (i.e. 3 hr post-stroke); however, the post-peak period was largely dampened in the 5d post-stroke signal. A similar alteration of the hemodynamic response function was also observed for the 5d post-stroke signal extracted from the Po nucleus when compared to the pre-stroke and 3 hr post-stroke signals (left panel, *Figure 4B*. Similar observations were made for the VPL).

Regarding the right hemisphere (i.e. non-ischemic; right panel, *Figure 4B*), the S1BF functional responses to left whisker stimulation were still reduced when compared with pre-stroke responses (black plot) but remained like the traces detected at 3 hr post-stroke (orange plot, non-significant). As for the left VPM, both the amplitude and time-to-peak of the hemodynamic responses from the right VPM responses were consistent with pre-stroke and 3 hr post-stroke values but the post-peak signal was decreased (brown plot). The functional responses extracted from the Po and VPL did not show changes when compared to pre-stroke and 3 hr post-stroke responses.

## Discussion

With this proof-of-concept study, we document on the feasibility of the continuous brain hemodynamics recording of a focal cerebral ischemia after MCAo in conscious rats. Using functional ultrasound imaging, we were able to extract multiple parameters (i.e. ischemia, location and spreading depolarization), characteristic of such cortical stroke. Then, we report on how the functional sensorimotor thalamocortical circuit was altered at early and late post-stroke stages.

Compared to highly-invasive conventional strategies such as clipping or suturing (*Macrae, 2011*; *Fluri et al., 2015*), the $FeCl_3$ model used here, is well suited to study stroke under awake conditions. Indeed, the use of $FeCl_3$ requires less manipulation, allows to maintain the dura intact, and strongly reduces the risk of hemorrhage (*Karatas et al., 2011*; *Syeara et al., 2020*) and animal loss. In addition, the $FeCl_3$ model closely mimics key features of human stroke, including focal ischemia, clot formation, minutes-long progressive occlusion of the vessel, possibility of vessel recanalization, and penumbral tissue (*Karatas et al., 2011*; *Syeara et al., 2020*). However, to adequately and efficiently occlude the vessel of interest, it is necessary to open the skull and to stabilize the cranial window under chronic conditions (i.e. case of delayed occlusion). It is worth noting that optimal preparation of the MCA window is highly critical, as the application of $FeCl_3$ is performed with a reduced field-of-view. In fact, as mentioned in the animal use report, one rat was excluded from the analysis due to spontaneous MCA re-perfusion, thus reducing the success rate of the model.

The $FeCl_3$-induced MCAo showed an abrupt and massive drop of blood perfusion remaining constant during the entire recording period. The ischemia was confined within the cortical territory perfused by the MCA (*Figure 2B*), and the infarct (location and size; *Figure 2G*) is in agreement with previous observations (*Syeara et al., 2020*; *Brunner et al., 2022a*). We also detected transient hyperemic events associated with spreading depolarizations (SDs) within the peri-ischemic territory,

with occurrence, frequency, and amplitude of the hemodynamic waves (*Figure 2D–F*) consistent with prior observation (*Brunner et al., 2022a*; *Nakamura et al., 2010*; *Takeda et al., 2011*; *Bere et al., 2014*; *Farkas et al., 2008*; *Farkas et al., 2010*; *Bogdanov et al., 2016*). Moreover, the spatiotemporal dynamic of the $FeCl_3$-induced MCAo is consistent with previous fUS imaging reports on cortical ischemia with various stroke models (*Brunner et al., 2018*; *Brunner et al., 2022a*; *Hingot et al., 2020*). In our awake stroke experimental context, cerebral hemodynamics and functional responses to stimuli in rats remain stable over the hour-long imaging sessions; however, we must mention the potential impact of such prolonged physical restraint on physiological and hemodynamic parameters (*Nagasaka et al., 1980*; *Aydin et al., 2011*; *Sikora et al., 2016*).

On top of tracking large hemodynamic variation (i.e. ischemia, SDs), one asset of the fUS imaging technology relies on its ability to track subtle hemodynamic changes in sparse brain regions (*Brunner et al., 2018*; *Macé et al., 2011*; *Urban et al., 2015*; *Macé et al., 2018*; *Brunner et al., 2020*; *Brunner et al., 2021*; *Urban et al., 2014*). Therefore, we have evaluated how evoked functional responses reorganize at early and late timepoints after stroke induction. Functional responses to mechanical whisker stimulation were detected in several regions relaying the information from the whisker to the cortex, including the VPM and Po nuclei of the thalamus, and S1BF, the somatosensory barrel-field cortex. Responses were also observed in the S2 cortex involved in the multisensory integration of the information (*Adibi, 2019*; *Bosman et al., 2011*; *Lohse et al., 2021*), the auditory cortex as it receives direct efferent projection from S1BF (*Zakiewicz et al., 2014*; *Lohse et al., 2021*), and the VPL nuclei of the thalamus are connected via corticothalamic projections (*Zakiewicz et al., 2014*).

Functional responses extracted in the left hemisphere affected by the focal ischemia (i.e. ipsilesional) show a primary alteration of the whisker-to-barrel pathway within the first hour after the stroke onset. While the abrupt loss in S1BF responses was mainly driven by the focal ischemia, the immediate but partial drop in VPM responses (*Figure 3D*) might result from the direct the loss of the excitatory corticothalamic feedback to the VPM (*Landisman and Connors, 2007*; *Bourassa et al., 1995*; *Temereanca and Simons, 2004*), or even from a dampening of thalamocortical excitability (*Tennant et al., 2017*). The absence of such cortical feedback suggests that the dampened functional responses might be driven by the intrinsic activity of the VPM in response to whisker stimulation. Five days after the initial injury, nuclei of the thalamus (VPM and Po) were subjected to a delayed and robust functional alteration (*Figure 4B*) as previously confirmed in other thalamic relay (*Tokuno et al., 1992*), probably associated with diaschisis, as previously characterized by tissue staining, reduction of metabolism, functions and perfusion (*Zhang et al., 2012*; *Carrera and Tononi, 2014*; *Cao et al., 2020*; *Viaene et al., 2011*; *Tokuno et al., 1992*). Functional responses of the S1BF extracted from the right hemisphere (i.e. contralesional) show a significant decrease shortly after the stroke onset (*Figure 3D*), and still detected at day 5, could be provoked by a loss of transcortical excitability (*Rema and Ebner, 2003*; *Li et al., 2005*). The late drop in VPM responses might be explained by corticothalamic modulation of the projections toward VPM (*Adibi, 2019*; *Li et al., 2005*).

While preliminary, these results obtained from awake head-fixed rats are in contradiction with a similar work by our group (fUS imaging, distal MCAo with microvascular clip, electrical whisker stimulation) showing higher contralesional responses to whisker stimulation during early stages of ischemic stroke (*Brunner et al., 2018*). However, these experiments were subjected to a long-term isoflurane regimen (surgery and imaging) known to alter functional responses (*Sicard et al., 2006*; *Paasonen et al., 2016*; *Ayata et al., 2004*) as well as disrupting hemodynamics (*Martin et al., 2006*). Therefore, further studies will be needed to accurately dissect the complex and long-lasting post-stroke alterations of the functional whisker-to-barrel pathway, including at the neuronal level by direct electrophysiology recordings and imaging, as fUS only report on hemodynamics as a proxy of local neuronal activity (*Sieu et al., 2015*; *Macé et al., 2018*; *Urban et al., 2014*; *Aydin et al., 2020*; *Sans-Dublanc et al., 2021*; *Nunez-Elizalde, 2021*). Another limitation relies on the experimental condition as our brain imaging paradigm was constrained to a single cross-section, thus missing out-of-plane brain regions also affected by the stroke (e.g. ischemic size, infract extension, origin, and propagation pattern of SDs)(*Topchiy et al., 2009*) or involved in the whisker network (e.g. superior colliculus, striatum, amygdala and cerebellum)(*Adibi, 2019*). To overcome such limitation, one can extend the size of the cranial window to allow for larger scale imaging either by sequentially scanning the brain (*Sieu et al., 2015*; *Macé et al., 2018*; *Brunner et al., 2021*; *Brunner et al., 2023*; *Hingot et al., 2020*; *Sans-Dublanc et al., 2021*; *Brunner et al., 2017*; *Brunner et al., 2022b*), or by using the recently

developed volumetric fUS which provides whole-brain imaging capabilities in anesthetized rats (*Rabut et al., 2019*) and awake mice/rats (*Brunner et al., 2020*). Finally, it is important to note that this proof-of-concept work did not specifically focus on the impact of (i) sex dimorphism, (ii) sleep/wake cycle on the stroke, or (iii) early behavioral outcomes following the insult that would greatly enhance the translational value of such preclinical stroke study (*Fisher et al., 2009*).

Beyond studying the whisker-to-barrel somatosensory circuit, the brain-wide capability of fUS opens the door to investigate on stroke-affected brain circuits and functions using transgenic lines combined with opto-/chemo-genetic strategies as the technology is fully mature for mice studies (*Macé et al., 2018*; *Brunner et al., 2020*; *Brunner et al., 2021*; *Sans-Dublanc et al., 2021*).

## Materials and methods

### Animals

The experimental procedures were approved by the Committee on Animal Care of the Katholieke Universiteit Leuven (ECD P172/2018), following the national guidelines on the use of laboratory animals and the European Union Directive for animal experiments (2010/63/EU). The manuscript was written according to the ARRIVE Essential 10 checklist for reporting animal experiments (*Percie du Sert et al., 2020*). Adult male Sprague-Dawley rats weighed between 250–400 g (n=9; Janvier Labs, France) were used. During habituation rats were housed two per cage and kept in a 12 hr dark/light cycle at 23 °C with ad libitum access to water and controlled access to food (15 g/rat/day). After the initial surgical procedure, rats were housed alone. See *Supplementary file 1* reporting on animal use, experimentation, inclusion/exclusion criteria.

### Body restraint and head fixation

The body restraint and head fixation procedures are adapted from published protocols and setups dedicated to brain imaging of awake rats (*Martin et al., 2006*; *Topchiy et al., 2009*; *Martin et al., 2002*). Rats were habituated to the workbench and to be restrained in a sling suit (Lomir Biomedical inc, Canada) by progressively increasing restraining periods from minutes (5 min, 10 min, 30 min) to hours (1 and 3 hr) for one or two weeks. The habituation to head-fixation started by short (5–30 s) and gentle head-fixation of the headpost between fingers. The headpost was then secured between clamps for fixation periods progressively increased following the same procedure as with the sling. For both body restraint and head fixation, the initial struggling and vocalization diminished over sessions. Habituation was completed when the rat remains still and calm over long restraint periods as previously established (*Topchiy et al., 2009*). Water and food gel (DietGel, ClearH2O, USA) were provided during all body restraint and head-fixation habituation sessions. Once habituated, the cranial window for imaging was performed as described below (*Figure 1A–C*).

### Surgical procedures

Cranial window over the MCA: Rats were anesthetized with isoflurane (5% for induction, 2% for maintenance; Iso-Vet, 1000 mg/g, Dechra, Belgium) and fixed in a stereotaxic frame. The depth of anesthesia was confirmed by the absence of reflex during paw pinching. After scalp removal and tissue cleaning, a 1 mm (*Fluri et al., 2015*) cranial window was performed at coordinates bregma +2 mm and lateral 7 mm, over the left distal branch of the MCA as reported in *Brunner et al., 2018*. A silicone plug (Body Double-Fast Set, Smooth-on, Inc, USA) was used to protect the window and ease the access to the MCA before the occlusion procedure. Then, a stainless-steel custom-designed headpost was fixed with bone screws (19010–00, FST, Germany) and dental cement (Super-Bond C&B, Sun Medical Co., Japan) to the animal skull (*Figure 1B*, left) as previously described by *Brunner et al., 2020*.

Cranial window for imaging: After recovery and habituation to head-fixation, a second cranial window was performed between bregma –2 to –4 mm and 6 mm apart from the sagittal suture (same anesthesia settings as the first cranial window; see above) following the procedure described in *Brunner et al., 2021*; *Figure 1B*, right. This cranial window aims to cover bilateral thalamocortical circuits of the somatosensory whisker-to-barrel pathway. A silicone plug was also used to protect the window and a headshield was added to secure it (*Urban et al., 2015*).

For both cranial windows, the dura mater was kept intact. After each surgery, rats were placed in their home cage and monitored until they woke up. Rats were medicated with analgesic (Buprenorphine,

0.1 mg/kg, Ceva, France), anti-inflammatory (Dexamethasone, 0.5 mg/kg, Dechra, Belgium) drugs injected directly after the surgery, at 24 hr and 48 hr after the surgery. An antibiotic (Emdotrim, 5%, Ecuphar, The Netherland) was added to the water bottle.

## Positioning

The mechanical fixation of the head-post ensures an easy and repeatable positioning of the ultrasound probes across imaging sessions. The ultrasound probe is indeed fixed to a micromanipulator enabling light adjustments. To find the plane of interest (containing both S1BF and thalamic relays: bregma -3.4 mm), we used brain landmarks (e.g. surface of the brain, hippocampus, superior sagittal sinus, large vessels). Note that as the headpost was carefully placed in the same position relative to skull landmarks (bregma and lambda), the position of the region of interest was minimal across animals.

## Chemo-thrombotic stroke induction with ferric chloride solution

Once the body were restrained and head-fixed the silicone plug covering the MCA window was removed allowing the application of a drop of 20% ferric chloride solution (FeCl$_3$; Sigma Aldrich, USA) to the MCA (*Karatas et al., 2011*; *Syeara et al., 2020*; *Figure 2*). Once the ischemia was visually detected using the real-time display of μDoppler images, the solution was washed out with saline to stop the reaction.

## Whisker stimulation paradigm

Two stimulation combs individually controlled by a stepper motor (RS Components, UK) were used to deliver mechanical 5 Hz sinusoidal deflection of ~20° of amplitude for 5 s, alternatively to left and right whisker pads. For each whisker pad, trials were spaced by a period of 1 min and 20 s without stimulation. Thus, the effective delay between two stimulations delivered to the same whisker pad is 80 s from start to start. The blocks of stimulation were continuously delivered throughout the imaging sessions, time-locked with the fUS acquisition (*Figure 3*) to allow the subsequent analysis of hemodynamic responses within the fUS time-series.

## Functional ultrasound imaging acquisition

Coronal μDoppler images were acquired using a 15-MHz linear probe composed of 128 piezo-elements spaced by 100 μm (L22-14Vx, Vermon, France) connected to a dedicated ultrasound scanner (Vantage 128, Verasonics, USA) and controlled by a high-performance computing workstation (fUSI-2, AUTC, Estonia). This configuration allowed us to image the brain vasculature with a resolution of 100 μm laterally, 110 μm in depth, and 300 μm in elevation (*Brunner et al., 2021*). The ultrasound sequence generated by the software is adapted from *Macé et al., 2018* and *Brunner et al., 2021* Ultrafast images of the brain were generated using five tilted plane-waves (–6°, –3°, +0.5°, +3°, +6°). Each plane wave is repeated six times, and the recorded echoes are averaged to increase the signal-to-noise ration. The five plane-wave images are added to create compound images at a frame rate of 500 Hz. To obtain a single vascular image we acquired a set of 250 compound images in 0.5 s, an extra 0.3 s pause is included between each image to have some processing time to display the images for real-time monitoring of the experiment. The set of 250 compound images has a mixed information of blood and tissue signal. To extract the blood signal, we apply a low pass filter (cut off 15 Hz) and an SVD filter that eliminates 20 singular values. This filter aims to select all the signals from blood moving with an axial velocity higher than ~1 mm/s. To obtain a vascular image we compute the intensity of the blood signal i.e., Power Doppler image. This image is in first approximation proportional to the cerebral blood volume (*Macé et al., 2011*; *Montaldo et al., 2022*). Overall, this process enables a continuous acquisition of power Doppler images at a frame rate of 1.25 Hz during several hours. Then, the acquired images are processed with a dedicated GPU architecture, displayed in real-time for data visualization, and stored for subsequent off-line analysis (*Brunner et al., 2021*).

## fUS data processing and analysis

The data processing was performed following the procedure described by *Brunner et al., 2021*.

### Registration to Paxinos rat brain atlas and data segmentation

We registered the fUS dataset to a custom digital rat brain atlas used in *Brunner et al., 2023*, using one coronal plane (bregma –3.4 mm) from the stereotaxic atlas of *Paxinos, 2014*. The image of the

brain vasculature was manually translated and rotated to align with the coronal plane of the reference atlas. For an accurate registration, we used landmarks such as the surface of the brain, hippocampus, superior sagittal sinus, and other large vessels. If needed, the brain volume was scaled to fit the atlas outline. The outcome of this registration procedure is an affine coordinate transformation: $\vec{r}\,' = M\vec{r} + \vec{a}$ , where $\vec{r} = (x, y, z)$ are the original coordinates image of the brain vasculature, M is the rotation and scaling matrix and $\vec{a}$ the translation vector. The dataset was segmented into 69 anatomical regions/hemispheres of the reference atlas (see *Supplementary file 2*). The hemodynamic signals were averaged in each area. The segmentation and the data processing were performed using an automated MATLAB-based pipeline. The software for data registration and segmentation is available in open-access (*Brunner et al., 2021*).

## Relative cerebral blood volume (rCBV)

We used the relative cerebral blood volume (rCBV, expressed in % as compared to baseline) to analyze ischemia, transient hemodynamic events associated with SDs and functional changes. rCBV is defined as the signal in each voxel compared to its average level during the baseline period. After registration and segmentation, the rCBV signal was averaged in each individual regions.

## Analysis of stroke hemodynamics

The extraction of the temporal traces from the ischemic area was performed based on the temporal analysis of the rCBV signal in the primary somatosensory barrel-field cortex (S1BF). The detection of hemodynamic events associated with SDs was performed based on the temporal analysis of the rCBV signal in the retrosplenial granular (RSGc) and dysgranular (RSD) cortices of the left hemisphere (ipsilesional). Hemodynamic events associated with SDs were defined as transient increase of rCBV signal (+25%) detected with a temporal delay of <10 frames (i.e. 8 s) between the two regions of interest, validating both the hyperemia and spreading features of hemodynamic events associated with spreading depolarizations (*Brunner et al., 2023*; *Bere et al., 2014*; *Ayata and Lauritzen, 2015*; *Binder et al., 2022*). This procedure allowed us to measure the occurrence of hemodynamic events associated with SDs over the recording period. Live recording of ischemia and spreading depolarizations can be visualized in *Video 1*.

## Activity maps

Pre- and post-stroke recordings are reshaped in 40 s sessions, i.e., 50 frames, centered on the start of the stimulation (at 20 s), and averaged based on the whisker stimulation paradigm (left or right). In each voxel, we compared signals along the recording in a time window before the stimulus onset and a time window after stimulus onset using a two-tailed Wilcoxon rank sum test. We obtained the z-statistics of the test for each voxel, and consequently a z-score for the coronal cross-section. Mean activity maps for left or right whisker stimulation (*Figures 3B and 4A*) show z-score value calculated using a Fisher's transform for all voxels across the coronal cross-section. Only voxels with a z-score >1.6 were considered significantly activated (p<0.05 for a one-tailed test).

## Hemodynamic response time-courses

The relative hemodynamic time course ΔrCBV was computed for each brain regions (after registration and segmentation; *Figures 3C–D , and 4B*), as the rCBV change compared to baseline at each time point. No additional filtering was used, and no trial was removed from the analysis.

## Statistical analysis

Activated brain regions were detected from hemodynamic response time-courses using GLM followed by t-test across animals as proposed in Brunner, Grillet et al., (*Brunner et al., 2021*). The area under the curve (AUC) from hemodynamic response time-courses was computed for individual trials in S1BF, VPM, and Po regions, for all the periods of the recording and for all rats included in this work. AUC were compared and analyzed using a non-parametric Kruskal-Wallis test corrected for multiple comparison using a Dunn's test. Tests were performed using GraphPad Prism 10.0.1.

## Histopathology

Rats were killed 24 hr after the occlusion for histological analysis of the infarcted tissue. Rats received a lethal injection of pentobarbital (100 mg/kg i.p. Dolethal, Vetoquinol, France). Using a peristaltic pump, they were transcardially perfused with phosphate-buffered saline followed by 4% paraformaldehyde (Sigma-Aldrich, USA). Brains were collected and post-fixed overnight. 50 μm thick coronal brain sections across the MCA territory were sliced on a vibratome (VT1000S, Leica Microsystems, Germany) and analyzed using the cresyl violet (Electron Microscopy Sciences, USA) staining procedure (see Open Lab Book for procedure). Slices were mounted with DPX mounting medium (Sigma-Aldrich, USA) and scanned using a bright-field microscope.

## Acknowledgements

The authors thank the members of the Fondation Leducq network #15CVD02, Dr. M Grillet, T Lambert and lab members for their insightful comments and discussions. We thank NERF animal caretakers, including I Eyckmans, F Ooms, and S Luijten, for their help with the management of the animals. *Figures 1–3* use BioRender.com icons. Funding This work is supported by grants from the Fondation Leducq (15CVD02) and KU Leuven (C14/18/099-STYMULATE-STROKE). The functional ultrasound imaging platform is supported by grants from FWO (MEDI-RESCU2-AKUL/17/049, G091719N, and 1197818 N), VIB Tech-Watch (fUSI-MICE), Neuro-Electronics Research Flanders TechDev fund (3D-fUSI project).

## Additional information

### Funding

| Funder | Grant reference number | Author |
| --- | --- | --- |
| Fonds Wetenschappelijk Onderzoek | G0C9923N | Alan Urban |
| Fonds Wetenschappelijk Onderzoek | G079623N | Alan Urban |
| Fonds Wetenschappelijk Onderzoek | 12D7523N | Clément Brunner |
| ERANET, EU Horizon 2020 | Grant number 964215, UnscrAMBLY | Alan Urban Gabriel Montaldo |

The funders had no role in study design, data collection and interpretation, or the decision to submit the work for publication.

### Author contributions

Clément Brunner, Conceptualization, Data curation, Formal analysis, Validation, Investigation, Visualization, Methodology, Writing – original draft, Writing – review and editing; Gabriel Montaldo, Data curation, Software, Formal analysis, Supervision, Writing – review and editing; Alan Urban, Conceptualization, Software, Supervision, Funding acquisition, Project administration, Writing – review and editing

### Author ORCIDs

Clément Brunner ⬡ https://orcid.org/0000-0002-2567-4832
Alan Urban ⬡ https://orcid.org/0000-0002-6460-2364

### Ethics

The experimental procedures were approved by the Committee on Animal Care of the Katholieke Universiteit Leuven (ECD P172/2018), following the national guidelines on the use of laboratory animals and the European Union Directive for animal experiments (2010/63/EU).

Reviewer #1 (Public Review): https://doi.org/10.7554/eLife.88919.3.sa1
Reviewer #2 (Public Review): https://doi.org/10.7554/eLife.88919.3.sa2

Reviewer #3 (Public Review): https://doi.org/10.7554/eLife.88919.3.sa3
Author Response https://doi.org/10.7554/eLife.88919.3.sa4

## Additional files

### Supplementary files

• Supplementary file 1. Reporting on animal use, experimentation, exclusion criteria, and figure association.

• Supplementary file 2. List of the 69 brain regions/hemispheres from the coronal cross-section µDoppler imaged in each rat organized by main anatomical structures. Adapted from the Paxinos rat brain atlas (*Paxinos, 2014*).

• MDAR checklist

### Data availability

All data generated or analyzed during this study is available online at https://doi.org/10.5281/zenodo.10074382.

The following dataset was generated:

| Author(s) | Year | Dataset title | Dataset URL | Database and Identifier |
|-----------|------|---------------|-------------|-------------------------|
| Brunner C, Montaldo G, Urban A | 2023 | Functional ultrasound imaging of stroke in awake rats | https://doi.org/10.5281/zenodo.10074382 | Zenodo, 10.5281/zenodo.10074382 |

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
