## [Editor Report · eLife assessment]

This **important** proof-of-concept study strongly supports the utility of functional ultrasound imaging for evaluating cerebral hemodynamics in rat models of brain injury. Functional ultrasound affords a distinct coverage/spatial/temporal resolution tradeoff when compared to other modalities for studying brain hemodynamics. The **solid** data presented indicate high fidelity of the recordings, a particular feat given that the rats were awake. On the other hand, single slice imaging and complexity of registration of subsequent imaging sessions limit the usefulness of the approach, particularly for quantitative imaging, and the small sample size will need to be followed up with and verified by future studies. This work will be of interest to researchers working in functional neuroimaging and more precisely with preclinical models of stroke in rodents.

---

## [Referee Report · Reviewer #1 (Public Review)]

Summary: The authors apply a new approach to monitor widespread changes in sensory evoked hemodynamic activity after focal stroke in fully conscious rats. Using functional ultrasound (fUS), they report immediate and lasting (up to 5 days) depression of sensory evoked responses in somatosensory thalamic and cortical regions.

Strengths: This a technically challenging study that employs new methods to study more distributed changes in sensory evoked neural activity, inferred from changes in cerebral blood flow. The authors provide compelling images and rigorous analysis to support their conclusions.

The primary weakness of this paper was the small sample size used for drawing conclusions. The authors have added additional references that help support their preliminary findings.

Ultimately, it is a proof of concept paper showing the potential of this imaging approach for examining brain wide changes in activity before and after stroke in awake animals. In that sense, I think this paper will be well appreciated by researchers trying to understand how stroke leads to distributed changes in brain function.

---

## [Referee Report · Reviewer #2 (Public Review)]

Brunner et al. present a new and promising application of functional ultrasound (fUS) imaging to follow the evolution of perfusion and haemodynamics upon thrombotic stroke in awake rats. The authors leveraged a chemically induced occlusion of the rat Medial Cerebral Artery (MCA) with ferric chloride in awake rats, while imaging with fUS cerebral perfusion with high spatial and temporal resolution (100µm x 110µm x 300µm x 0.8s). The authors also measured evoked haemodynamic responses at different timepoints following whisker stimulation.

As the fUS setup of the authors is limited to 2D imaging, Brunner and colleagues focused on a single coronal slice where they identified the primary Somatosensory Barrel Field of the Cortex (S1BF), directly perfused by the MCA and relay nuclei of the Thalamus: the Posterior (Po) and the Ventroposterior Medial (VPM) nuclei of the Thalamus. All these regions are involved in the sensory processing of whisker stimulation. By investigating these regions the authors present the hyper-acute effect of the stroke with these main results:

- MCA occlusion results in a fast and important loss of perfusion in the ipsilesional cortex.

- Thrombolysis is followed by Spreading Depolarisation measured in the Retrosplenial cortex.

- Stroke-induced hypo-perfusion is associated with a significant drop in ipsilesional cortical response to whisker stimulation, and a milder one in ipsilesional subcortical relays.

- Contralesional hemisphere is almost not affected by stroke with the exception of the cortex which presents a mildly reduced response to the stimulation.

In addition, the authors demonstrate that their protocol allows to follow up stroke evolution up to five days postinduction. They further show that fUS can estimate the size of the infarcted volume with brilliance mode (Bmode), confirming the presence of the identified lesional tissue with post-mortem cresyl violet staining.

Upon measuring functional response to whisker stimulation 5 days after stroke induction, the authors report that:

- The ipsilesional cortex presents no response to the stimulation

- The ipsilesional thalamic relays are less activated than hyper acutely

These observations mainly validate a new method to chronically image the longitudinal sequelae of stroke in awake animals. However, the potentially more intriguing results the authors describe in terms of functional reorganization of functional activity following stroke will require additional data to be validated. While highly preliminary, the research model proposed by the author (where the loss of the infarcted cortex induces reduces activity in connected regions, whether by cortico-thalamic or cortico-cortical loss of excitatory drive), is interesting. This hypothesis would require a greatly expanded, sufficiently powered study to be validated (or disproven)."

---

## [Referee Report · Reviewer #3 (Public Review)]

The authors set out to demonstrate the utility of functional ultrasound for evaluating changes in brain hemodynamics elicited acutely and subacutely by middle cerebral artery occlusion model of ischemic stroke in awake rats.

Functional ultrasound affords a distinct set of tradeoffs relative to competing imaging modalities. Acclimatization of rats for awake imaging has proven difficult with most, and the high quality of presented data in awake rats is a major achievement. The major weakness of the approach is in its being restricted to single slice acquisitions, which also complicates registration of acquisition across multiple imaging sessions within the same animal. Establishing that awake imaging represents an advancement in relation to studies under anesthesia hinges upon establishment of the level of stress experienced by the animals in the course of imaging, i.e., requires providing data on the assessment of stress over the course of these long imaging sessions, which was not undertaken. This is particularly significant given that physical restraint has been established to be a particularly potent stressor in experimental models of stress. Assessment of the robustness of these measurements in a larger cohort of animals under varying conditions is of particular significance for supporting its wide applicability.

---

## [Author Response]

The following is the authors’ response to the original reviews.

Thank you for your consideration and insightful comments on our article.

We have gone through all the reviewers' comments and addressed all their questions and concerns point by point.

As per their recommendation, we have amended our manuscript by providing more information about the experimental procedure and statistical analysis followed, and removed some analyses with a reduced number of imaging sessions. In addition, as a Resource and Tools article, the claim of our paper has been adjusted to a proof-of-concept paper showing robust and reliable preliminary results. In the meantime, we have provided 3 new Supplementary Figures, including one showing data from all individual animals.

**Reviewer #1 (Public Review):**
The authors apply a new approach to monitor brain-wide changes in sensory-evoked hemodynamic activity after focal stroke in fully conscious rats. Using functional ultrasound (fUS), they report immediate and lasting (up to 5 days) depression of sensory-evoked responses in somatosensory thalamic and cortical regions.Strengths: This a technically challenging and proof-of-concept study that employs new methods to study brain-wide changes in sensory-evoked neural activity, inferred from changes in cerebral blood flow. Despite the minor typos/grammatical errors and small sample size, the authors provide compelling images and rigorous analysis to support their conclusions. Overall, this was a very technically difficult study that was well executed. I believe that it will pave the way for more extensive studies using this methodological approach. Therefore I support this study and my recommendations to improve it are relatively minor in nature and should be simple for the authors to address.Weaknesses: The primary weakness of this paper is the small sample sizes. Drawing conclusions based on the small sham control group (n=2) or 5-day stroke recovery group (n=2), is rather tenuous. One way to alleviate some uncertainty with regard to the conclusions would be to state in the discussion that the findings (ie. loss of thalamocortical function after stroke) are perfectly consistent with previous studies that examined thalamocortical function after stroke. The authors missed some of these supporting studies in their reference list (see PMID: 28643802, 1400649). A second issue that can easily be resolved is their analysis of the 69 brain regions. This seems like a very important part of the study and one of the primary advantages of employing efUS. As presented, I had difficulty seeing the data. I think it would be worthwhile to expand Fig 3 (especially 3C) into a full-page figure with an accompanying table in the Supplementary info section describing the % change in CBF for each brain region.Other Recommendations for the authors:.Since there is variability in spreading depolarizations, was there any trend in the relationship between # SD's and ischemic volume? I know there are few data points but a scatterplot might be of interest.For statistical comparisons of 'response curves' in Fig 3 and 4, what exactly was the primary dependent measure: changes in peak amplitude (%) or area under the curve?There are several typos and minor grammatical errors in the manuscript. Some editing is recommended.

We thank the reviewer for the comments and suggestion, we have adapted our message to a proof-of-concept paper showing robust and reliable preliminary results. We also thank the reviewer for pointing out important references that support our observation and have added them to our article. We have provided a supplementary full-page version of the current Figure 3C (see Supplementary Figure 3).

Regarding the recommendations, we strongly agree that it would be of interest to link SDs and ischaemia, but unfortunately this can't be done because our experimental design, i.e. narrow cranial window and single static plane, does not allow brain-wide quantification of ischemic volume. This would be possible either by scanning the brain or by using a matrix array (also discussed in the manuscript).

For statistical analysis of the hemodynamic response curves, we have adapted them to compare the area under the curve (AUC). In addition, we have provided a new Supplementary Figure 4 showing the associated values and statistics.

We have edited typos and errors.

**Reviewer #2 (Public Review):**
Brunner et al. present a new and promising application of functional ultrasound (fUS) imaging to follow the evolution of perfusion and haemodynamics upon thrombotic stroke in awake rats. The authors leveraged a chemically induced occlusion of the rat Medial Cerebral Artery (MCA) with ferric chloride in awake rats, while imaging with fUS cerebral perfusion with high spatio and temporal resolution (100µm x 110µm x 300µm x 0.8s). The authors also measured evoked haemodynamic response at different timepoints following whisker stimulation.As the fUS setup of the authors is limited to 2D imaging, Brunner and colleagues focused on a single coronal slice where they identified the primary Somatosensory Barrel Field of the Cortex (S1BF), directly perfused by the MCA and relay nuclei of the Thalamus: the Posterior (Po) and the Ventroposterior Medial (VPM) nuclei of the Thalamus. All these regions are involved in the sensory processing of whisker stimulation. By investigating these regions the authors present the hyper-acute effect of the stroke with these main results:MCA occlusion results in a fast and important loss of perfusion in the ipsilesional cortex.Thrombolysis is followed by Spreading Depolarisation measured in the Retrosplenial cortex.Stroke-induced hypo-perfusion is associated with a significant drop in ipsilesional cortical response to whisker stimulation, and a milder one in ipsilesional subcortical relays.Contralesional hemisphere is almost not affected by stroke with the exception of the cortex which presents a mildly reduced response to the stimulation.In addition, the authors demonstrate that their protocol allows to follow up stroke evolution up to five days post-induction. They further show that fUS can estimate the size of the infarcted volume with brilliance mode (B-mode), confirming the presence of the identified lesional tissue with post-mortem cresyl violet staining.Upon measuring functional response to whisker stimulation 5 days after stroke induction, the authors report that:The ipsilesional cortex presents no response to the stimulationThe ipsilesional thalamic relays are less activated than hyper acutelyThe contralesional cortex and subcortical regions are also less activated 5d after the stroke.These observations mainly validate the new method as a way to chronically image the longitudinal sequelae of stroke in awake animals. However, the potentially more intriguing results the authors describe in terms of functional reorganization of functional activity following stroke appear to be preliminary, and underpowered ( N = 5 animals were imaged to describe hyper-acute session, and N = 2 in a five day follow-up). While highly preliminary, the research model proposed by the author (where the loss of the infarcted cortex induces reduces activity in connected regions, whether by cortico-thalamic or cortico-cortical loss of excitatory drive), is interesting. This hypothesis would require a greatly expanded, sufficiently powered study to be validated (or disproven).

We thank the reviewer for the careful and accurate description of our work. We have addressed all the comments, recommendations and concerns raised by providing details of the experimental procedure and statistical analysis followed, and by removing some analyses associated with a reduced number of imaging sessions (at d5, n=2).

**Reviewer #3 (Public Review):**
The authors set out to demonstrate the utility of functional ultrasound for evaluating changes in brain hemodynamics elicited acutely and subacutely by the middle cerebral artery occlusion model of ischemic stroke in awake rats.Functional ultrasound affords a distinct set of tradeoffs relative to competing imaging modalities. Acclimatization of rats for awake imaging has proven difficult with most, and the high quality of presented data in awake rats is a major achievement. The major weakness of the approach is in its being restricted to single-slice acquisitions, which also complicates the registration of acquisition across multiple imaging sessions within the same animal. Establishing that awake imaging represents an advancement in relation to studies under anesthesia hinges upon the establishment of the level of stress experienced by the animals in the course of imaging, i.e., requires providing data on the assessment of stress over the course of these long imaging sessions. This is particularly significant given how significant a stressor physical restraint has been established to be in rodent models of stress. Furthermore, assessment of the robustness of these measurements is of particular significance for supporting the wide applicability of this approach to preclinical studies of brain injury: the individual animal data (effect sizes, activation areas, kinetics) should thus be displayed and the statistical analysis expanded. Both within-subject, within/across sessions, and across-subjects variability should be evaluated. Thoughtful comments on the relationship between power doppler signal and cerebral blood volume are important to include and facilitate comparisons to studies recording other blood volume-weighted signals. Finally, the contextualization of the observations with respect to other studies examining acute and subacute changes in brain hemodynamics post focal ischemic stroke in rats is needed. It is also quite helpful, for establishing the robustness of the approach, when the statistical parametric maps are shown in full (i.e. unmasked).

We would like to thank the reviewer for the comments, recommendations and concerns he/she/they raised. We have addressed all the points to clarify our article and make it more relevant and informative for readers.

**Reviewer #2 (Recommendations For The Authors):**
The work described by Brunner et al is primarily a methodological paper, with potentially interesting, yet not robust enough, novel biological insight into the mechanisms of stroke. Nonetheless, the method employed is interesting and potentially well-validated.General comments/suggestions1- One potential concern I have is related to the relatively low sample size used, with n=5 for the main results and only n=2 for the follow-up after 5d. I am not sure much can be generalized using only two animals in any research study and this N = 2 dataset should probably be removed entirely from the study. Moreover, I found the statistical methods used were only superficially described, which prevented me from assessing whether the results reported by the authors are biologically relevant or not (including some signiﬁcant differences in rCBV well below 1% estimated over two individuals).

We fully agree with the reviewer’s comment and balanced our claim by considering this work as a proof-of-concept on brain imaging of multiple aspects of stroke hemodynamics (ischemia, spreading depolarization-like events, cortico-thalamic functions) in awake head-fixed rats. Therefore, we attenuated our message along the entire manuscript to prevent misunderstanding and over statement (e.g., Lines 356, 441, 455), we also remove statistics from the analysis at d5 post-stroke, see Figure 4 and associated paragraph from Line 356.

2- Based on their investigations, the authors propose a model where the loss of infarcted cortex induces reduced activity in connected regions, whether by cortico-thalamic or cortico-cortical loss of excitatory drive. This is an intriguing framework but this hypothesis would require a more complete, well-powered study to be substantiated.I think a clear recognition of the fact that these findings are just preliminary and not validated should be more explicitly reported. I also marginally note here that these results are in contrast with previous reports from the same team where occlusion of the MCA induced increased response to whisker stimulation in anaesthetised rats. These contradictory findings are not discussed in this manuscript.

As mentioned above, we explicit more on the proof-of-concept proposed in this work as well as clearly stating on the preliminary aspect of the findings described in this work. As mentioned above, we attenuated our message along the entire manuscript to prevent misunderstanding and over statement (e.g., Lines 348, 433, 447), we also remove statistics from the analysis at d5 post-stroke, see figure 4 and associated paragraph from Line 348.

We thanks the reviewer for pointing out the missing link with our previous work performed under anesthesia. We therefore provided a discussion point on this contradictory finding (Line 441).

3- In a previous study from the same group perfusion was imaged in 3D either by means of a motorized probe or by using a 2D matrix arrays. It would be interesting to discuss why a 2D approach was chosen in this study over those previous methods.

Indeed, brain-wide coverage would be of great interest in such experiment context. As mentionned by the reviewer, two strategies can be used:

• One can scan the brain using a motorized probe as performed for different purposes by Sieu et al., Nature Methods, 2015; Hingot, Brodin et al., Theranostics 2020; Macé et al., Neuron 2019 and also by our group in Sans-Dublanc, Chrzanowska et al., Neuron, 2022; Brunner et al. Frontiers in Neuroscience 2022 and Brunner et al., JCBFM 2023. (This list of publication is not exhaustive).

• A second approach aims at using a 2D matrix array to capture functions at brain-wide scale. So far, this strategy has been employed in a couple of studies (Rabut et al., Nature Methods, 2019 and Brunner, Grillet et al., Neuron, 2020).

The strategy consisting of scanning (manually or using a motor) strongly limits investigation on brain functions, as performing an accurate covering of the functional regions requires an extensive and time-consumming scanning: brain functions must be addressed several time to capture a reliable and robust signal for all the brain section scanned (see Brunner et al., 2022). Unfortunately, this strategy prevents us to accurately capture other brain hemodynamics like the dynamic of the ischemia or the spreading depolarization event.

On the other hand, the volumetric functional ultrasound imaging (vfUSI) would be suited for brain-wide coverage capturing large-scale brain functions (see Brunner, Grillet et al. Neuron 2020) and hemodynamic events (see Rabut et al., Nature Methods, 2019) but at the cost of the resolution, frame rate and larger cranial window. Unfortunately, this technology was not available when this work was conducted.

Such experimental opportunities have been suggested at the end of the manuscript: “To overcome such limitation, one can extend the size of the cranial window to allow for larger scale imaging either by sequentially scanning the brain27,28,31,32,59,69,71,72, or by using the recently developed volumetric fUS which provides whole-brain imaging capabilities in anesthetized73 and awake rats30.“

4- Overall the registration scheme seems suboptimal which ultimately questions the speciﬁcity of the ﬁndings in thalamic regions. It would be interesting to validate this procedure, especially the probe repositioning five days after the stroke.

Positioning was not difficult part of this experiment. First, all head posts were implanted in the same position relative to the skull references bregma and lambda. Second, the head fixation ensures the same placement of the headpost for all animals. Finally, fine adjustement of the ultrasound probe position were done using a micromanipulator by finding key landmarks from the µDoppler image. In practice, minimal adjustements were needed to find back the same imaging plane. We provide additional information about the positionning in the Materials and Methods section.

New text – Line 126: “Positionning.

The mechanical fixation of the head-post ensures an easy and repeatabe positionning of the ultrasound probe across imaging session. The ultrasound probe is indeed fixed to a micromanipulator enabling light adjustements To find the plane of interest (containing both S1BF and thalamic relays: bregma - 3.4mm), we used brain landmarks (e.g., surface of the brain, hippocampus, superior sagittal sinus, large vessels). Note that as the headpost was carefully placed in the same position relative to the skulls landmarks (bregma and lambda), the position of the region of interest was minimal across animals.”

Second, at d5 post-stroke, we positionned the ultrasound probe over the imaging window as described in the Materials and Methods section and use brain landmarks from baseline/post-stroke image to maximize the position of brain image. We better detail the procedure followed.

Original text: “First, we used the vascular markers and the shape of the hippocampus31,32 to find back the coronal cross-section imaged during the pre-stroke session. Five days after the MCA occlusion,….”

New text – Line 360 :“Five days after the MCA occlusion, we first placed the ultrasound probe over the imaging window and adjusted its position (using micromanipulator) to find back the recording plane from Pre-Stroke session using Bmode (morphological mode) and µDoppler imaging using brain vascular landmarks (i.e., vascular patterns, brain surface and hippocampus34,35; see Figure 2B).”

More detailed questions/comments/suggestionsMethodsARRIVE methodologyPoint 2b: sample size is not adequately explained, especially the use of n = 2 animals for 5d follow up

We have explicited the sample size by adding a short paragraph at the beginning of the Results section. We also make the Supplementary Table 1 more accurate.New text – Line 239: “Animals

Report on animal use, experimentation, exclusion criteria can be found in Supplementary Table 1. Rat#1 was excluded after the control session as the imaging window was too anterior to capture both cortical and thalamic responses. Ra#2 was excluded as hemodynamic responses were inconsistent during baseline (pre-stroke) period. Rat#3 showed early post-stroke reperfusion and was excluded from stroke analysis, the control session (pre-stroke) from Rat#3 was analyzed.”

Point 7: statistical methods: The quantiﬁcation used to assess signiﬁcant differences in stimulation traces is poorly described.

We have amended the Materials and Methods section about statistics and provided Supplementary Figure 4.

New text – Line 221: “Activated brain regions were detected from hemodynamic response time-courses using GLM followed by t-test across animals as proposed in Brunner, Grillet et al.,34. The area under the curve (AUC) from hemodynamic response time-courses was computed for individual trials in S1BF, VPM and Po regions, for all the periods of the recording and for all rats included in this work. AUC were compared and analysed using a non-parametric Kruskal-Wallis test corrected for multiple comparison using a Dunn’s test. Tests were performed using GraphPad Prism 10.0.1. “

Functional Ultrasound Imaging acquisitionReferences 26 and 28 imply 2.5Hz and 2Hz acquisition rates, respectively. Why does the same method result in a 1.25Hz acquisition rate here? Can you conﬁrm the same spatial resolution in these conditions?

The spatial resolution is independent of the temporal resolution (frame rate). The spatial resolution depends on the resolution of the compound image and the temporal resolution is given by the number of compound images to generate a single Doppler image (exposure time). By increasing the number of compound images, the frame rate decreases while increasing the signal to noise ratio and sensistivity. For some work, a pause between 2 frames is used (mostly due to technical limitations in the software (processing time , or execution of a real-time display/processing by the user)), however this reduces the frame rate.

**Author response table 1. sa4table1:** Comparing with the sequences used in references 26 and 28, we have the following timing parameters.

	Sequence	Frequency (Hz)	N Compound	Acquisition time (s)	Pause (s)	Frame rate (Hz)
Macé et al.,	High sensibility	500	200	0.4	0.2	1.7
Neuron, 2018	High speed	500	50	0.1	0.1	10
Brunner et al.,	High sensibility	500	250	0.5	0	2
Nat Prot, 2021	High speed	500	50	0.1	0	10
This study		500	250	0.5	0.3	1.25

In this work, we decided to reduce the frame rate to have less images but with higher SNR. The 0.3s were added by technical considerations in this specific implementation.

New text – Line 158:“ To obtain a single vascular image we acquired a set of 250 compound images in 0.5s, an extra 0.3s pause is included between each image to have some processing time to display the images for real-time monitoring of the experiment. “

Activity MapsHow is the use of a 40s window motivated?

The 40s window has been choosen to better compare hemodynamic responses to either left or right whisker stimulation and centered the period of interest on the start of the stimulation.Original text:” Pre- and post-stroke recordings are reshaped in shorter 40-s sessions, i.e., 50 frames, …”

New text – Line 206:“ Pre- and post-stroke recordings are reshaped in 40-s sessions, i.e., 50 frames, centered on the start of the stimulation (at 20s), …”

I think the manuscript would benefit from the use of an established, event-based GLM for activity mapping.

We thank the reviewer for this suggestion, here we used a z-score for activity mapping that is largerly established in the neuroimaging realm.

The statistical thresholds used should account for multiple comparisons.

We have amended the Materials and Methods section, and figure captions about statistics and provided Supplementary Figure 4.

Statistical analysesOverall this section is only superficially described, and lacks detailed information.

We have amended the Materials and Methods section about statistics and provided Supplementary Figure 4.

New text – Line 221 : “Activated brain regions were detected from hemodynamic response time-courses using GLM followed by t-test across animals as proposed in Brunner, Grillet et al.,34. The area under the curve (AUC) from hemodynamic response time-courses was computed for individual trials in S1BF, VPM and Po regions, for all the periods of the recording and for all rats included in this work. AUC were compared and analysed using a non-parametric Kruskal-Wallis test corrected for multiple comparison using a Dunn’s test. Tests were performed using GraphPad Prism 10.0.1. “

Are average rCBV changes referred to in the 40s window?

The rCBV changes are referring to the pre-stimulation baseline. We have modified the text accordingly (Line 206).

Were normality and variance equality requirements verified in the group with n=2?

Based on reviewers comment’s on the limited amount of recording at 5d, we have decided to remove this statistical analysis. The manuscript, figure and caption were corrected accordingly.

There is no method for cresyl violet staining

We thank the review for highlighting this omission. We have provided a paragraph in the Materials & Methods section detailling the histology procedure – Line 228:

“Histopathology

Rats were killed 24hrs after the occlusion for histological analysis of the infarcted tissue. Rats received a lethal injection of pentobarbital (100mg/kg i.p. Dolethal, Vetoquinol, France). Using a peristaltic pump, they were transcardially perfused with phosphate-buffered saline followed by 4% paraformaldehyde (Sigma-Aldrich, USA). Brains were collected and post-fixed overnight. 50-μm thick coronal brain sections across the MCA territory were sliced on a vibratome (VT1000S, Leica Microsystems, Germany) and analyzed using the cresyl violet (Electron Microscopy Sciences, USA) staining procedure (see Open Lab Book for procedure). Slices were mounted with DPX mounting medium (Sigma-Aldrich, USA) and scanned using a bright-field microscope.”

Results 1: Real time imaging of stroke induction in awake ratsWhy is the window so narrow in the anteroposterior direction?

The imaging window was defined based on the brain regions investigated in this work, meaning the primary somatosensory cortex (S1BF) and the ventroposterior medial thalamic relay (VPM). From Paxinos atlas, a position of interest is located at Bregma -3.4mm. The cranial window was performed accordingly, and restricted couple of mm to avoid non-needed procedure and brain exposure.We added a new sentence in the Materials & Methods section – Line 116: “This cranial window aims to cover bilateral thalamo-cortical circuits of the somatosensory whisker-to-barrel pathway.”

What validation was employed for the habituation protocol? Are animals stressed by the procedure? Do you have cortisol data to show? Ar animal weights throughout the procedure?

The habituation protocol employed in this work follows recommandations from the expert in the field and peers (Martin et al., Journal of Neuroscience Methods, 2002; Martin et al., Neuroimage 2006; Topchiy et al., Behav Brain Res 2009). We have amended the corresponding paragraph in the Materials & Methods section detailling the habituation procedure:

Original text: “Body restraint and head fixation.

Rats were habituated to the workbench and to be restrained in a sling suit (Lomir Biomedical inc, Canada), progressively increasing the restraining period from minutes to hours33,34. After the headpost implantation (see below), rats were habituated to be head-fixed while restrained in the sling. The period of fixation was progressively increased from minutes to hours. Water and food gel (DietGel, ClearH2O, USA) were provided along the habituation session. Once habituated, the cranial window for imaging was performed as described below (Figure 1A-C).”

New text - Line 90:“ Body restraint and head fixation.

The body restraint and head fixation procedures are adapted from published protocols and setup dedicated for brain imaging of awake rats39–41. Rats were habituated to the workbench and to be restrained in a sling suit (Lomir Biomedical inc, Canada) by progressively increasing restraining periods from minutes (5mins, 10mins, 30mins) to hours (1 and 3hrs) for one or two weeks. The habituation to head-fixation started by short (5 to 30s) and gentle head-fixation of the headpost between fingers. The headpost was then secured between clamps for fixation periods progressively increased following the same procedure as with the sling. For both body restraint and head fixation, the initial struggling and vocalization diminished over sessions. Water and food gel (DietGel, ClearH2O, USA) were provided for all body restraint and head-fixation habituation sessions. Once habituated, the cranial window for imaging was performed as described below (Figure 1A-C).”

The observation of contralateral oligemia is based only on RSG traces.

We provided contralesional perfusion changes for all regions in Supplementary Figure 1.

The spatial and temporal distribution of Bmode measured hyperechogenicity is surprising and should be discussed. Reference 29 describes for instance non-overlap with an area of hypo-perfusion. Overlap between hypo-perfused and infarct volumes should be systematically investigated and coregistered with histology. Moreover, reference 40, while using a different model, presents hyperechogenicity at 5h.

The B-mode images in Figure 2B are presented as an illustration of the potential morphological changes detected at different timepoint. However, our study focuses on functional responses and not on the evolution of the morphological changes. Indeed, this Bmode images remain difficult to interpret as they show a structural reorganization at the level of the ultrasound scatterers which has not been directly linked with tissue infarction, oedema, orother histological conditions.

Regarding the reference 40, the authors found an hyper-echogenicity at 5h a time window is not covered by our protocol. In reference 29, we indeed detailed a mismatch between the µDoppler images and histopathology. As suggested by the reviewer, seeking for other potential mismatchs/overlaps between Bmode/µDoppler and histopathology is an interesting field on investigation, but remains out of the scope of this work.

Results 3: Delayed alteration of the somatosensory thalamocortical pathwayThese results are underpowered and as such should probably be removed entirely from the paper (or substantiated with greater Ns of animals).Based on reviewers comment’s on the limited amount of recording at 5d, we have decided to remove this statistical analysis. The manuscript, figure and caption were corrected accordingly.If I am not mistaken, reference 28 describes a protocol for awake mouse imaging, and thereby does not introduce any hippocampal landmark allowing effective positioning of the probe.

We thanks the reviewer for this comment. While not used in the figure detailling image registration in reference 28, step 42 (page 17) from the protocol mentions the use of hippocampal landmark to position of the imaged brain to the atlas. The hippocampal landmark is also used in Brunner et al., JCBFM 2023, we have added this reference which is more appropriate to this work (i.e., rat model, digitalized paxinos atlas, linear ultrasound transducer).

Signiﬁcant difference in ispsilesional VPM with post-stroke period looks spurious.

We have amended the Materials and Methods section about statistics and provided Supplementary Figure 4.

Discussion:The sentence "might result from the direct loss of the excitatory corticothalamic feedback to the VPM" should be moderated in the absence of electrophysiology support. Such a decrease could be explained by reduced perfusion due to the challenge.

The reviewer is right and we believe the tense used in the sentence already balance the claim. However, we clarified on how such result could be better validated.

Original text: “Further work will need to dissect the complex and long-lasting post-stroke alterations of the functional whisker-to-barrel pathway, including at the neuronal level, as fUS only reports on hemodynamics as a proxy of local neuronal activity27,28,60,66–68“

New text – Line 445: “Therefore, further studies will be needed to accurately dissect the complex and long-lasting post-stroke alterations of the functional whisker-to-barrel pathway, including at the neuronal level by direct electrophysiology recordings and imaging, as fUS only reports on hemodynamics as a proxy of local neuronal activity30,31,63,74–76.“

Figure 2Panel B would be more informative if presented as an average.

The aim of this figure is to show the raw data of a typical case. Averaging µDoppler images wouldn’t be illustrative as individual vessels will not be visible anymore. Because the vessels are in different positions from one animal to another, an average image would be blurred.

Panel C lacks contralateral S1BF trace.

We have provided contralesional perfusion changes for all regions in Supplementary Figure 1.

Methods for detection of SDs refer to non-peer-reviewed reference 29, where SD is deﬁned as 50% over baseline level. What is the actual threshold/method used to deﬁne a SD in this study?

We better detailled this procedure in the Materials & Methods section - Line 195: “The detection of hemodynamic events associated with spreading depolarizations (SDs) was performed based on the temporal analysis of the rCBV signal in the retrosplenial granular (RSGc) and dysgranular (RSD) cortices of the left hemisphere (ipsi-lesional). SDs were defined as transient increase of rCBV signal (+25%) detected with a temporal delay of <10 frames (i.e., 8secs) between the two regions of interest, validating both the hyperemia and spreading features of hemodynamic events associated with spreading depolarizations.”

For panel F, a measure of variance would be more suited to show stereotypic proﬁle across animals as the number of SDs varies between animals.

Figure 2F indeed shows the average profile of hemodynamic events associated with spreading depolarizations (black line) with the variance (95% confidence interval error bands in gray). We have adjusted the corresponding figure caption to make this information more clear.

Figure 3The exact stimulation employed is not clear as the methods describe a 1.33 min delay between two whisker pad stimulations, but the figure reports 40s. The description is thereby ambiguous.We thank the reviewer for pointing out this potiential confusion which allowed us to correct a mistake

• The effective delay between two stimulations delivered to the whisker pads is 40 seconds

• The effective delay between two stimulations delivered to the same whisker pad is 80 seconds from start to start or 75 seconds from end to start.

The text was amended accordingly in line 144: “Thus, the effective delay between two stimulations delivered to the same whisker pad is 80 seconds from start to start.“

In panel B the choice of colormap and transparency for template overlay is not explained and is confusing given the employed threshold of 1.6. Which mask was used to overlay the activation map on the template? Why black color to represent a supposedly signiﬁcant difference?

We thank the reviewer for pointing out this potiential confusion. We have adjusted the colormap in Figures 3 and 4.

The pre-stroke thalamic response is clearly localized in VPM for left stimulation, while it overlaps VPM and Po for the right stimulation. This questions the accuracy of the employed registration scheme and consequently the choice of these ROIs, which appear quite small as compared to the resolution and this positioning precision.

We see the point of the reviewer, here the apparent difference because the brain is slighly tilted. By adjusting the angle for both activity maps (see Author response image 1) we confirm that both maps are very similar including the for activated areas VPM and Po.

**Author response image 1. sa4fig1:** 

It would be interesting to see the same activation maps for all animals in supplementary.

We have provided the Supplementary Figure 5 that contains both ipsilateral and contralateral responses to whiskers stimulation (from both left and right pads) for all trials and all rats included in this work.

Looking at panel C, more cortical regions seem to respond to the stimulation above S1BF.

The reviewer is right and we have indeed mentioned this point several times in the original manuscript in:

• the result section: “We also detected significant increase of activity in S2, AuD, Ect (****p<0.0001) and PRh (***p<0.001) cortices and VPL nucleus (**p<0.01; the list of acronyms is provided in Supplementary Table 2), brain regions receiving direct efferent projections from the S1BF45,48,49, VPM or Po nuclei50–52.”

• the caption of Figure 4: “S1BF, S2, AuD, VPM, VPL and Po regions are brain regions significatively activated all pvalue<0.01; GLM followed by t-test.”

• the conclusion section : “Functional responses to mechanical whisker stimulation were detected in several regions relaying the information from the whisker to the cortex, including the VPM and Po nuclei of the thalamus, and S1BF, the somatosensory barrel-field cortex. Responses were also observed in the S2 cortex involved in the multisensory integration of the information43,44,61, the auditory cortex as it receives direct efferent projection from S1BF45,61, and the VPL nuclei of the thalamus connected via corticothalamic projections45.“

It would be interesting to see bilateral traces as supplementary figures.

We have provided the Supplementary Figure 5 that contains both ipsilateral and contralateral responses to whiskers stimulation (from both left and right pads) for all trials and all rats included in this work.

In both panels C and D, n=5 is reported, but methods state the use of 7 animals. Please clarify how animals have been used in the different studies

We have clarified the report on animal use and amended the Supplementary Table 1 accordingly.

In Panel D, the 95% CI intervals seem particularly narrow. Might this be the result of considering multiple trials as independent events? A GLM analysis would avoid this statistical fallacy.

We have provided the Supplementary Figure 5 that contains both ipsilateral and contralateral responses to whiskers stimulation (from both left and right pads) for all trials and all rats included in this work. The statistical analysis has been adjusted (see Materials and Methods) and completed with a Supplementary Figure 4

Figure 4See comments above for Figure 3

We have adjusted the Figure 3 accordingly to reviewer’s suggestions

**Reviewer #3 (Recommendations For The Authors):**
1. Introduction: Given the emphasis on the awake state, it would be helpful to note that a significant portion of strokes occur during sleep - as well as comment on its hemodynamic difference with respect to an awake state.

We agree with the reviewer on the remark that some strokes occur during sleep phase. However, here the awake state, which has been poorly addressed in the litterature, is opposed to anesthesia a condition largerly used to investigate brain functions after stroke. We added a point and corresponding references about wake-up stroke, see Line 49.

1. The effects of anesthetics on stroke are quite variable and the literature data on the topic is rather divergent: it would be helpful for the introduction to reflect the large level of discord in the literature and the wide-ranging mechanisms of action of different anesthetics.

We thank the reviewer for this comment. We have completed our original sentence in the introduction to better reflect the various effects of anesthetics on stroke, see Line 50

1. The reference list (14-17) to other studies of brain hemodynamic changes post ischemic stroke is egregiously short. Please expand. Similarly, the list of citations to other functional ultrasound rodent studies in the literature (23-24) is misleading: other groups have published similar work and ought to be cited.

We thank the reviewer for this comment and added complementary references.However, we believe that the references 14-17 pointed by the reviewer are not only refering to brain hemodynamic changes but mostly on network and function as stated in the manuscript. Regarding references on fUS (23-24) mentioned by the reviewer, we did not limited our citation on functional ultrasound imaging to those 2 articles but on 15+ from 4 different research groups.

1. It would be helpful if the authors used "spreading depolarization" the way it has been utilized in the many decades of research on them in the literature, namely, as waves of hyper/hypoactivity in the electrophysiological signals. Please use a distinct term to refer to waves of changes in the hemodynamic state.

We have amended the terminology used in the manuscript. “Spreading depolarization” has been replaced by “hemodynamic events associated with spreading depolarizations” or similar.

1. Why is this investigation restricted to male rats?

As a proof of concept, we did not performed experiments in female rats. We agree that further investigation would require a gender mix. We added a line in the discussion.

New text – Line 455:” Finally, it is important to note that this proof-of-concept work did not specifically focus the impact of sex dimorphism on the stroke or early behavioral outcomes following the insult that would greatly enhance the translational value of such preclinical stroke study80.”

1. Were the animals tested during their active phase? If not, why not, and what are the implications of testing their responses during the sleep phase?

We think there is a misunderstanding here as we investigated brain functions in awake head-fixed rats. Therefore, the sleep/active phases were not investigated neither mentioned in the manuscript.

1. How is the level of stress monitored/established?

In this work, we followed established procedure used to reduce stress and disconfort of the rats all along the experiment. The procedure used is now better detailled in the Materials and Methods section. However, the level of stress was not monitored, and would be of interest to considere in future experiments.

1. What are the sequelae of stress on brain hemodynamics, especially given 1-4 hour long sessions.

This is a good remark. While we cannot state on how the stress impacts brain hemodynamics, the data extracted show that hemodynamics reponse functions were stable and robust over hour-long recording (see control and pre-stroke sessions in Supplementary Figure 5).

1. How is the animal prepared for stroke induction? In general, the methodological steps surrounding animal handling and preparation are exceedingly terse.

We provided more details about the handling and preparation of the rats in the Materials and Methods section.

Original text: “Body restraint and head fixation.

Rats were habituated to the workbench and to be restrained in a sling suit (Lomir Biomedical inc, Canada), progressively increasing the restraining period from minutes to hours33,34. After the headpost implantation (see below), rats were habituated to be head-fixed while restrained in the sling. The period of fixation was progressively increased from minutes to hours. Water and food gel (DietGel, ClearH2O, USA) were provided along the habituation session. Once habituated, the cranial window for imaging was performed as described below (Figure 1A-C).”

New text - Line 90:“ Body restraint and head fixation.

The body restraint and head fixation procedures are adapted from published protocols and setup dedicated for brain imaging of awake rats39–41. Rats were habituated to the workbench and to be restrained in a sling suit (Lomir Biomedical inc, Canada) by progressively increasing restraining periods from minutes (5mins, 10mins, 30mins) to hours (1 and 3hrs) for one or two weeks. The habituation to head-fixation started by short (5 to 30s) and gentle head-fixation of the headpost between fingers. The headpost was then secured between clamps for fixation periods progressively increased following the same procedure as with the sling. For both body restraint and head fixation, the initial struggling and vocalization diminished over sessions. Water and food gel (DietGel, ClearH2O, USA) were provided for all body restraint and head-fixation habituation sessions. Once habituated, the cranial window for imaging was performed as described below (Figure 1A-C).”

1. What is the reproducibility of the chemo-thrombotic model timeline? What are its limitations?

We have provided more information on the chemo-thrombotic model and its limitations in the discussion section to discuss

New text – Line 402:” However, to adequatly and efficiently occlude the vessel of interest, removing a piece of skull remains required. As mentioned in the report on animal use, one rat was excluded from the analysis as the MCA spontaneously reperfuses, thus dropping the success rate of such model.”

1. What is the motivation behind the 5-days post stroke timepoint selection?

In addition to demonstrating the feasability of imaging brain functions at different timepoint following the ischemia, the motivation to performed this delayed session was to capture functional diaschisis which is known to occur few days after the initial insult. More recurrent imaging sessions covering a longer post-stroke period would be of high interest to better capture the impact of ischemia on both the brain hemodynamics and functions.

1. How predictive is hyperacute hemodynamics imaging of the long-term outcome?

We thanks the reviewer for this question, that remains of major interest in the stroke realm. However, the prediction of long-term outcome would require to capture brain hemodynamic at larger scale as performed in Hingot et al., Theranostics 2020 and Brunner et al. JCBFM 2023, a coverage not accessible with the imaging window proposed in this work.

1. It would be greatly reassuring if the authors presented the statistical parametric maps without masking regions of interest (eg Fig3B).

We thank the reviewer for pointing out this potential confusion. In the first version of the figure, the colormap used of activity maps was indeed non optimal. Therefore, we (i) adjusted the colormap used in Fig 3 and 4 and (ii) provided non-thresholded z-score maps for all rats in Supplementary Figure 5.

1. Fig 3C is hard to make out.

We provided a full page version of the Figure 3C in Supplementary Figure 3.

1. Figs 3,4 should incorporate box and whisker plots of data across all rats scatter plots of individual animal data.

We are not sure which kind of data the reviewer wants to have displayed here.However, we have provided the Supplementary Figure 5 that contains both ipsilateral and contralateral responses to whiskers stimulation (from both left and right pads) for all trials and for individual animal included in this work.

1. The final panels in Figures 3,4 would more tellingly include the plots of the linear models fitted.

Based on all reviewers’ comments, we have adjusted and clarified the statistical analysis performed (see Materials and Method) and completed with a Supplementary Figure 4.

1. The frame rate calculations are not adding up unless averaging and pauses are included so some more details should be stated. Are tilted plane waves averaged before compounding as in prior publications?

The angles are averaged 6 times before compounding to reduce signal to noise ration and there is a pause of 0.3s between each Doppler image. See also question “Functional Ultrasound Imaging acquisition” from reviewer 2. We also provided supplementary and key information about the sequence used in this work.

We have provided complementary information in the manuscript:

Original text:” The ultrasound sequence generated by the software is the same as in Macé et al.,26 and Brunner, Grillet et al., Briefly, the ultrafast scanner images the brain 140 with 5 tilted plane-waves (-6°, -3°, +0.5°, +3°, +6°) at a 10-kHz frame rate. The 5 plane-wave images are added to create compound images at a frame rate of 500Hz. Each set of 250 compound images is 142 filtered to extract the blood signal. Finally, the intensity of the filtered images is averaged to obtain a 143 vascular image of the rat brain at a frame rate of 1.25Hz. Then, the acquired images are processed with a dedicated GPU architecture, displayed in real-time for data visualization, and stored for subsequent off-line analysis.”

New text – Line 146:” The ultrasound sequence generated by the software is adapted from Macé et al.31 and Brunner, Grillet et al.34 Ultrafast images of the brain were generated using 5 tilted plane-waves (-6°, -3°, +0.5°, +3°, +6°). Each plane wave is repeated 6 times and the recorded echoes are averaged to increase the signal to noise ration. The 5 plane-wave images are added to create compound images at a frame rate of 500Hz. To obtain a single vascular image we acquired a set of 250 compound images in 0.5s, an extra 0.3s pause is included between each image to have some processing time to display the images for real-time monitoring of the experiment. The set of 250 compound images has a mixed information of blood and tissue signal. To extract the blood signal we apply a low pass filter (cutt off 15Hz) and an SVD filter that eliminates 20 singular values. This filter aims to select all the signal from blood moving with an axial velocity higher than ~1mm/s. To obtain a vascular iimage we compute the intensity of the blood signal i.e., Power Doppler image. This image is in first approximation proportional to the cerebral blood volume26,28. Overall, this process enables a continious acquisition of power Doppler images at a frame rate of 1.25Hz during several hours.”

1. Ultrasound data processing: The filtering process should have more description. It would be highly instructive to explain that the power Doppler signal is being used and comment clearly on its relationship to blood volume, commenting on stalled flow mircrovessels/RBC-devoid micrrovessels, and considerations of vessel orientation.

The compound image has a mixed information of blood and tissu signal. To extract the blood signal, we applied a low pass filter (cutt off 15Hz) and an SVD filter that eliminates 20 singular values. This filter selects all the signal from blood moving with an axial velocity higher than ~1mm/s. To obtain a vascular iimage we compute the intensity of the blood signal (Power Doppler image). This power Doppler image is in first approximation proportional to the cerebral blood volume.

These information have been added in the Materials and Methods section of the manuscript.

1. Does the SVD processing have the same cut off (20 singular values) as in prior publications as a standard value, or is that adjusted for each study? There are enough minor differences between sequences that these details are uncertain. Do the overall hemodynamics measurements (Fig 2) include all data acquired, or do they exclude the whisker stimulation events, and if so, how long of a window is excluded? The explanation of the activity maps should be rephrased e.g. "... recordings are segmented in shorter 40-s time windows encompassing the whisker stimulation trials..."

We agree that these details are important, all these information have been added to the manuscript

• SVD processing: We eliminate 20 singular values as in cited studies.

• Sequence: we have included more details about the sequence.

• Processing: all data during the whisker stimulation is used.

• We have rephrased the explanation about the activity maps.

1. Discuss the methodology behind histological data shown in Fig. 1.

We thank the review for highlighting this omission. We have provided a paragraph in the Materials & Methods section detailling the histology procedure (Line 228):

“Histopathology

Rats were killed 24hrs after the occlusion for histological analysis of the infarcted tissue. Rats received a lethal injection of pentobarbital (100mg/kg i.p. Dolethal, Vetoquinol, France). Using a peristaltic pump, they were transcardially perfused with phosphate-buffered saline followed by 4% paraformaldehyde (Sigma-Aldrich, USA). Brains were collected and post-fixed overnight. 50-μm thick coronal brain sections across the MCA territory were sliced on a vibratome (VT1000S, Leica Microsystems, Germany) and analyzed using the cresyl violet (Electron Microscopy Sciences, USA) staining procedure (see Open Lab Book for procedure). Slices were mounted with DPX mounting medium (Sigma-Aldrich, USA) and scanned using a bright-field microscope